# Structural and functional consequences of NEDD8 phosphorylation

Katrin Stuber[1,2,3], Tobias Schneider[2,3], Jill Werner [1,2], Michael Kovermann [2,3✉], Andreas Marx [2,3✉] & Martin Scheffner [1,3✉]

Ubiquitin (Ub) and Ub-like proteins (Ubls) such as NEDD8 are best known for their function as covalent modifiers of other proteins but they are also themselves subject to post-translational modifications including phosphorylation. While functions of phosphorylated Ub (pUb) have been characterized, the consequences of Ubl phosphorylation remain unclear. Here we report that NEDD8 can be phosphorylated at S65 - the same site as Ub - and that S65 phosphorylation affects the structural dynamics of NEDD8 and Ub in a similar manner. While both pUb and phosphorylated NEDD8 (pNEDD8) can allosterically activate the Ub ligase Parkin, they have different protein interactomes that in turn are distinct from those of unmodified Ub and NEDD8. Among the preferential pNEDD8 interactors are HSP70 family members and we show that pNEDD8 stimulates HSP70 ATPase activity more pronouncedly than unmodified NEDD8. Our findings highlight the general importance of Ub/NEDD8 phosphorylation and support the notion that the function of pUb/pNEDD8 does not require their covalent attachment to other proteins.

[1] Dept. of Biology, University of Konstanz, Konstanz, Germany. [2] Dept. of Chemistry, University of Konstanz, Konstanz, Germany. [3] Konstanz Research School Chemical Biology, University of Konstanz, Konstanz, Germany. ✉email: michael.kovermann@uni-konstanz.de; andreas.marx@uni-konstanz.de; martin.scheffner@uni-konstanz.de

The covalent attachment of ubiquitin (Ub) and ubiquitin-like molecules (Ubls) to substrate proteins is one of the most prevalent post-translational modifications in eukaryotes. Aside from being a signal for proteasomal degradation, ubiquitylation is involved in the regulation of virtually every cellular process, including cell division, signal transduction, transport pathways as well as other forms of protein degradation including autophagy[1,2]. NEDD8 is with 57% sequence identity the closest relative of Ub and displays many of Ub's structural determinants, such as the conserved I44 hydrophobic patch[3]. NEDD8 is best known for its function as a stimulator of cullin-RING ligases (CRLs) via neddylation of cullin family proteins[4,5]. It also modifies many other proteins, though the consequences of neddylation remain in many cases unclear[3,6,7]. This is at least in part due to the notion that only little is known about NEDD8-specific interaction partners.

Various human disorders have been ascribed to dysregulation of proteins involved in assembling, binding, or disassembling Ub or Ubl conjugates[8,9]. In a juvenile form of Parkinson's disease, for instance, the gene encoding the RING-between-RING (RBR) Ub ligase Parkin is mutated such that the Ub ligase function is defective. Under normal conditions, Parkin is present in an auto-inhibited state that upon certain stress stimuli can be converted into an active state by phosphorylation of a distinct serine residue of its Ub-like domain. This structural transition is preceded by PINK1-mediated phosphorylation of Ub (pUb) at S65 (pS65). In turn, pUb binds to Parkin and activates Parkin by two mechanisms, by releasing the Ub-like domain resulting in its phosphorylation by PINK1 and by allosteric activation of Parkin[10–15]. Once activated, Parkin ubiquitylates mitochondrial proteins, and the Ub-decorated mitochondrion is recognized and removed by the autophagy machinery, a process termed mitophagy[16]. It was also reported that neddylation of Parkin and PINK1 can result in increased Parkin activity[17,18]. However, if NEDD8 can activate Parkin also in a non-covalent manner, has not been addressed so far.

NMR spectroscopic studies revealed that pUb adopts two conformations[19]. The major species resembles unmodified Ub with perturbations of chemical shifts in the vicinity of pS65 only, while the minor species shows significant structural changes. In the minor species, the C-terminal tail of Ub is retracted into the Ub core (Ub-CR), thereby extending the pS65-containing loop and disrupting the I44 hydrophobic patch[19]. The pH of the solution - and thus the protonation state of the phosphate group - influences the equilibrium between these two states[20,21], insofar as the double deprotonated phosphate group favors the C-terminally retracted form. Furthermore, the Ub-CR conformation is also found at low abundance in unmodified Ub, and PINK1 preferentially phosphorylates the retracted form of Ub indicating that phosphorylation stabilizes the retracted form rather than inducing it[22].

The role of pUb in mitophagy has been investigated in detail[16]. Yet, the possibilities that proteins apart from Parkin and PINK1 bind to pUb and/or to Ub-CR and that phosphorylation of Ub plays a role in other signaling pathways have not been addressed so far. Similarly, although NEDD8 also harbors a serine residue at position 65 (Fig. 1a) that can be phosphorylated[23–25], the functional and structural consequences of this phosphorylation remain elusive. Here we show that PINK1 phosphorylates NEDD8 at S65 (pNEDD8) in vitro and that pNEDD8 activates Parkin. NMR analysis of pNEDD8 revealed that there are significant structural changes upon S65 phosphorylation and that it exists in a conformational equilibrium between two different states. Similar to pUb, the second conformation shows β5-strand slippage that retracts the C-terminal tail into the core. By using genetic code expansion, we generated pUb and pNEDD8 and

non-hydrolyzable forms thereof and used these as bait molecules for affinity enrichment coupled to high-resolution mass spectrometry (AE-MS) approaches. By this, we identified selective interactors of pUb and pNEDD8 at a proteomic scale. Finally, we show that pNEDD8 allosterically stimulates the ATPase activity of HSP70, which we identified as a preferential interaction partner of pNEDD8.

## Results

**pNEDD8 activates Parkin.** Like Ub, NEDD8 harbors a serine residue at position 65 (Fig. 1a), and several proteomic studies have shown that S65 can be phosphorylated[23–25]. To obtain insight into the potential effects of this phosphorylation on NEDD8 structure and function, we first confirmed that as reported[14], NEDD8 can be phosphorylated by PINK1 and asked whether S65 is the actual site of phosphorylation. To do so, bacterially expressed NEDD8 and Ub were incubated with PINK1. Indeed, PINK1 phosphorylated NEDD8 at S65 in vitro (Fig. 1b, Supplementary Fig. 1) with an efficiency of approximately 50 percent of that of Ub phosphorylation.

The finding that PINK1 phosphorylates NEDD8 at S65 underscores the similarity between NEDD8 and Ub. Thus, we next asked whether similar to pUb[10], pNEDD8 increases the efficiency of PINK1-mediated phosphorylation of Parkin and in addition can act as an allosteric activator of the Ub ligase function of Parkin. To generate sufficient amounts of homogeneous populations of pNEDD8 and pUb, we employed the method of genetic code expansion to site-specifically incorporate phosphoserine in bacteria[26]. Quantitative incorporation of phosphoserine into NEDD8 and Ub as well as the homogeneity of the purified phosphorylated NEDD8 and Ub variants was confirmed by ESI-MS (Supplementary Fig. 1) and Phos-tag gel analysis (Supplementary Fig. 2). With pNEDD8 and pUb in hands, we first studied their effect on PINK1-mediated phosphorylation of Parkin. Here, we observed that pNEDD8 increased Parkin phosphorylation with an efficiency similar to pUb (Fig. 1c). Then, we determined the ability of pNEDD8 to allosterically activate Parkin in comparison to pUb. Full-length human Parkin showed basal activity in the presence of unmodified Ub which was stimulated by the addition of pUb (ratio Ub:pUb of 9:1) (Fig. 1d, Supplementary Fig. 2). Remarkably, the addition of pNEDD8 also led to significant activation of Parkin autoubiquitylation, with pNEDD8 being approximately 2-3-fold less efficient than pUb, whereas unphosphorylated NEDD8 had no effect (Fig. 1d). Moreover, a Parkin variant (Parkin H302A) with a mutation in the pUb binding interface was only poorly activated by pUb, as reported before[10,27], as well as by pNEDD8 (Fig. 1d). This indicates that pNEDD8 stimulates Parkin activity by direct interaction with the pUb binding site. To ensure that the observed effects do not require the covalent attachment of pNEDD8 and to be able to directly compare the effects of pNEDD8 and pUb on Parkin activity, we used forms of pNEDD8 and Ub that harbor a C-terminal extension (pNEDD8-His, pUb-His) and, thus, cannot be activated and conjugated to proteins (Fig. 1d, e). As shown in Fig. 1e, the addition of increasing amounts of pNEDD8-His resulted in dose-dependent activation of Parkin. The pUb-His positive control showed similar behavior with pUb-His being about 2-fold more efficient in activation than pNEDD8-His.

In line with the above findings, we could demonstrate that pNEDD8 binds to recombinant Parkin. As for pUb, the pNEDD8-Parkin interaction did not depend on the presence of Parkin's Ub-like domain, while the interaction with the pUb binding interface mutant Parkin H302A was significantly diminished (Fig. 1f). Furthermore, pNEDD8 bound somewhat less efficiently than pUb to Parkin which correlates with their

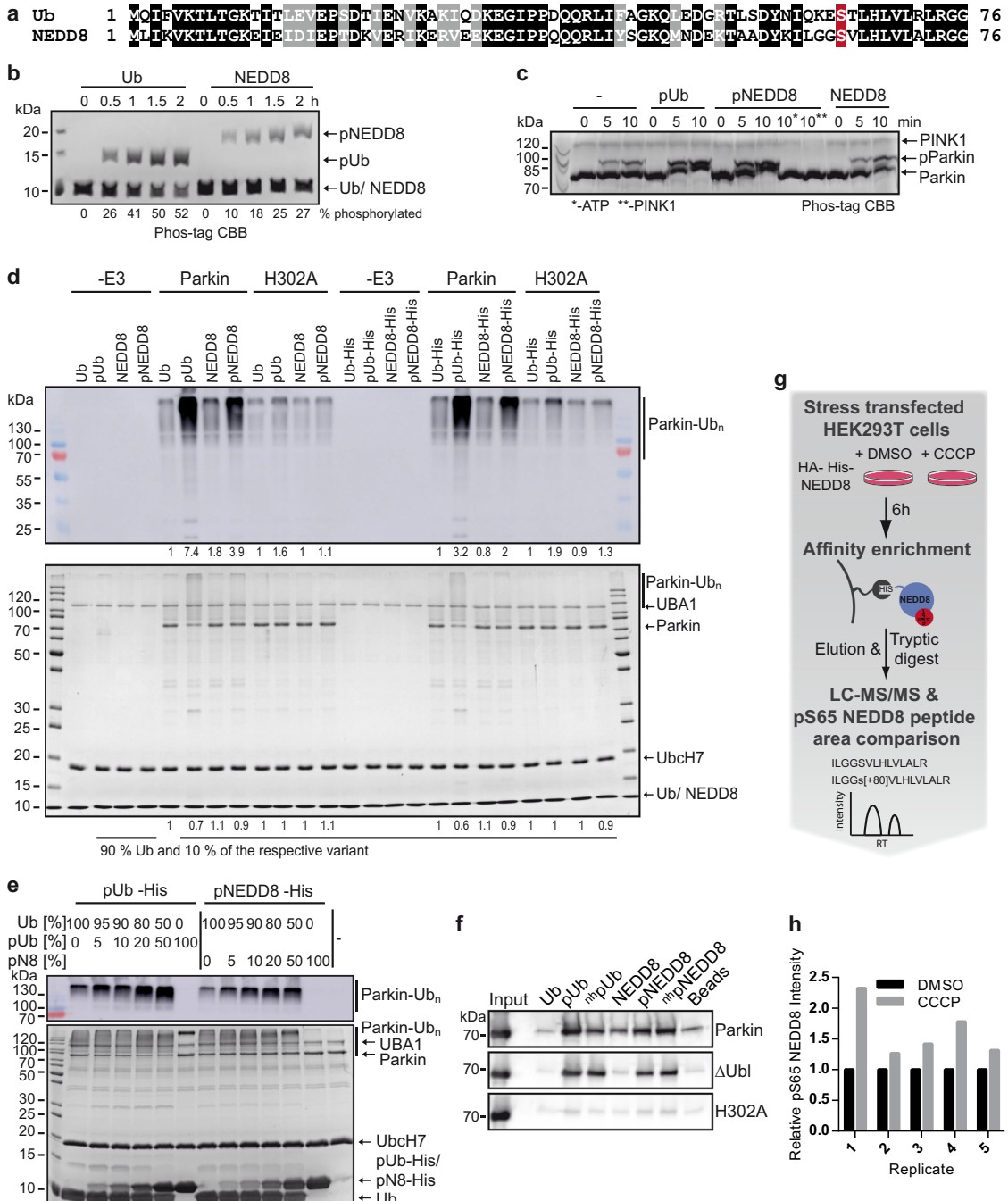

efficiency in Parkin activation (Fig. 1d, e). In conclusion, pNEDD8 can activate Parkin by activating its Ub ligase activity and by increasing the efficiency of PINK1-mediated Parkin phoshphorylation and, thus, represents an alternative or additional possibility for Parkin activation.

To determine, if similar to Ub, mitochondrial stress stimulates NEDD8 phosphorylation, we transfected HEK293T cells with an expression construct encoding NEDD8 with an N-terminal HA-His tag and induced mitochondrial depolarization by addition of CCCP (N-(4-chlorophenyl)carbonohydrazonoyl dicyanide). Upon affinity enrichment under denaturing conditions (Fig. 1g), NEDD8/neddylated proteins were subjected to tryptic digestion and the resulting peptides were analyzed in a targeted proteomics approach (Supplementary Fig. 1). This showed that upon mitochondrial stress, the relative abundance of the pS65-containing NEDD8 peptide was

elevated about 1.5-fold compared to the mock control (Fig. 1h, Supplementary Fig. 1), which appears to be similar to what was reported for the stimulation of Ub phosphorylation by mitochondrial depolarization[12].

**S65 phosphorylation of NEDD8 induces a conformational equilibrium between two structurally differing states.** Since the retracted conformation of Ub appears to be the preferred substrate for PINK1 and is stabilized by S65 phosphorylation[22], we wondered whether pNEDD8 also adopts two interchangeable conformations. To investigate the effect of S65 phosphorylation on the conformational dynamics of NEDD8, we generated isotopically labeled NEDD8 as well as pNEDD8 for high-resolution NMR spectroscopy. Comparing the corresponding two-dimensional heteronuclear

**Fig. 1 NEDD8 phosphorylated at S65 activates Parkin. a** Protein sequence alignment of Ub and NEDD8. Identical amino acids are indicated by a black background, similar amino acids by a grey background. The serine residue at position 65 is highlighted by a red background. **b** *Ph*PINK1 phosphorylates NEDD8. NEDD8 or Ub were incubated with *Ph*PINK1 for the times indicated. Reactions were subjected to Phos-tag SDS-PAGE followed by Coomassie blue staining. **c** pNEDD8 enhances Parkin phosphorylation by *Ph*PINK1. GST-Parkin was incubated with *Ph*PINK1 in the absence (−) or presence of either pUb, pNEDD8, or NEDD8 for the times indicated. Reactions were subjected to Phos-tag SDS-PAGE followed by Coomassie blue staining. *, the reaction in the absence of ATP; **, the reaction in the absence of *Ph*PINK1. **d, e** Parkin/Parkin H302A autoubiquitylation was performed as described in Methods in the presence or absence of the Ub/NEDD8 variants indicated. The ratios of the different Ub/NEDD8 variants present in the reactions are indicated in percent (in **d**, 90% corresponds to 35 µM Ub, 10% correspond to 4 µM of the Ub/NEDD8 variants indicated; in **e**, 100% corresponds to 39 µM of Ub or 30 µM of pUb-His/pNEDD8-His). Reactions were stopped after 60 min and analyzed by SDS-PAGE followed by Western blot analysis with an anti-Ub antibody (upper panels) or by Coomassie blue staining (lower panels). **d** Levels of the ubiquitylated forms of Parkin (upper panel) and of the non-modified form (lower panel) were quantified with respective levels in the presence of Ub set to 1. The fold change is indicated below the panels. **d, e** Running positions of molecular mass markers, unmodified Parkin, ubiquitylated forms of Parkin, UBA1, UbcH7, and Ub/NEDD8 variants are indicated. **f** The different Ub/NEDD8 variants were used as affinity matrices and incubated with GST fusion proteins of Parkin, a Parkin variant devoid of the Ub-like domain (ΔUbl) or the H302A mutant. Upon affinity enrichment, 25% of the respective elution fractions were subjected to SDS-PAGE followed by Western blot analysis with an anti-Parkin antibody. Input represents 5% of the respective Parkin variant. **g, h** Phosphorylation of S65 of NEDD8 under mitochondrial stress. **g** Schematic overview of the workflow for the identification of S65-phosphorylated NEDD8 peptides under normal growth conditions (DMSO) or upon mitochondrial stress (CCCP). **h** 80% of the HA-His-NEDD8 elution fraction were digested with trypsin, subjected to targeted LC-MS/MS and the normalized relative intensity of the pS65-containing NEDD8 peptide plotted. Shown are five biological replicates measured in technical duplicates (1, 2) or triplicates (3-5). See also Supplementary Fig. 1; source data are provided as a Source Data file.

$^1$H-$^{15}$N HSQC spectra, a considerably higher number of spectrally separated cross-peaks are observed with pNEDD8 than with NEDD8 (Fig. 2a, Supplementary Fig. 3). In brief, a total of 71 backbone amide resonances were assigned in the NEDD8 spectrum (Supplementary Fig. 3), whereas two independent sets with 62 and 59 coherent resonances, respectively, were identified in the pNEDD8 spectrum (Fig. 2a). In addition, two respective cross-peaks are present originating from D21, I36, and G47 that could not be ascribed to a distinct single set unambiguously. Thus, similar to the situation with pUb[19–21], two structurally different conformations of pNEDD8 exist in parallel.

In order to display the structural properties of the two conformations adopted by pNEDD8, we calculated weighted chemical shift perturbations (CSP) for each state of pNEDD8 compared to NEDD8 as well as between both states of pNEDD8 (Fig. 2b-d). The analysis revealed that both conformational states strongly differ from each other (Fig. 2d). However, one of them exhibits only little differences to unmodified NEDD8 which are mainly located next to the phosphorylation site (Fig. 2b), whereas the other state undergoes substantial structural changes (Fig. 2c). Since this is reminiscent to the situation with pUb, we apply the nomenclature originally introduced for pUb also to pNEDD8 and term the conformation that is similar to unmodified NEDD8 the 'relaxed state' and the other the 'retracted state'[21]. From relative signal heights of selected residues in the $^1$H-$^{15}$N HSQC NMR spectra, we estimate that both conformations are almost equally populated at pH 7.4 for pNEDD8 (45 ± 4% relaxed state) as well as for pUb (56 ± 4% relaxed state).

To further describe the kinetics of the conformational equilibrium of pNEDD8, we performed a ZZ exchange experiment that gives access to dynamics on the slow NMR time scale (Supplementary Fig. 4)[28]. Indeed, additional cross-peaks appeared in the corresponding spectra enabling the quantitation of the conformational exchange between both states of pNEDD8. This revealed an interconversion rate of 8.3 ± 0.3 s$^{-1}$ for pNEDD8 which is in the range of the number reported for pUb (2 s$^{-1}$)[19] indicating that S65 phosphorylation induces comparable structural and dynamic consequences in NEDD8 and Ub.

**The retracted state conformation of pNEDD8 shows β5-strand slippage.** Rearrangement of the hydrogen bonding network in the surrounding of S65 plays a key role in the formation of the retracted state conformation of pUb. It causes the retraction of the C-terminal β5-strand by two residues along with its alternating leucine (L67, L69, L71, L73) pattern[19,22]. Since S65 of NEDD8 is integrated in hydrogen bonds in the same way and the leucine pattern is also conserved in NEDD8, we wondered whether pNEDD8 undergoes a similar conformational change as pUb. Indeed, CSP values calculated between the retracted state of pNEDD8 and the relaxed state of unmodified NEDD8 are of significance especially in the region of the β5-strand as well as in the preceding loop and 3$_{10}$-helix (Fig. 2c). In addition, residues K4 and Y45, which are directly connected to S65 via hydrogen bonds, also exhibit CSP values with high amplitudes. This indicates that the region comprising the S65-containing loop and the adjacent secondary structure elements (3$_{10}$-helix and β5-strand) undergo a fundamental structural change accompanied by remodeling of the corresponding hydrogen bridges.

Focusing on this event, we performed secondary structure analysis on the basis of chemical shifts determined by TALOS-N[29] and compared the relaxed and retracted state conformations of pNEDD8 (Fig. 2e, Supplementary Fig. 5). Despite the large differences observed in the CSP mapping (Fig. 2d), only slight alterations of the secondary structure elements are apparent (Fig. 2e). Strikingly, however, in the retracted state the second 3$_{10}$-helix is elongated by one residue (K60) and the β5-strand is shifted towards the C terminus by one residue (from V66-L71 to L67-A72) in comparison to the relaxed state. These are principally the same differences that emerged from a structural comparison between the NMR solution structures obtained for both conformations of pUb[21]. Here, the second 3$_{10}$-helix is extended by three residues and the β5-strand is moved up by two residues in the retracted state conformation of pUb. In fact, in a complementary analysis based on chemical shift indices (CSI) we noted that a shift of the β5-strand by one residue (from S65-V70 to V66-L73) can also be proposed for the retracted state conformation of pNEDD8 (Supplementary Fig. 5). A closer look at the torsion angles (ψ and φ) derived from TALOS-N further confirms that the β5 slippage occurs in pNEDD8, since besides K60 the strongest deviations between both conformational states are mainly associated with residues in the leucine pattern mentioned above (L67, L71, L73) (Supplementary Fig. 5). In conjunction with the CSP data, we thus demonstrate that the conformational change characterized by retraction of the C-terminal β5-strand is not limited to pUb and at low abundance to unmodified Ub[19,22], but is also present in

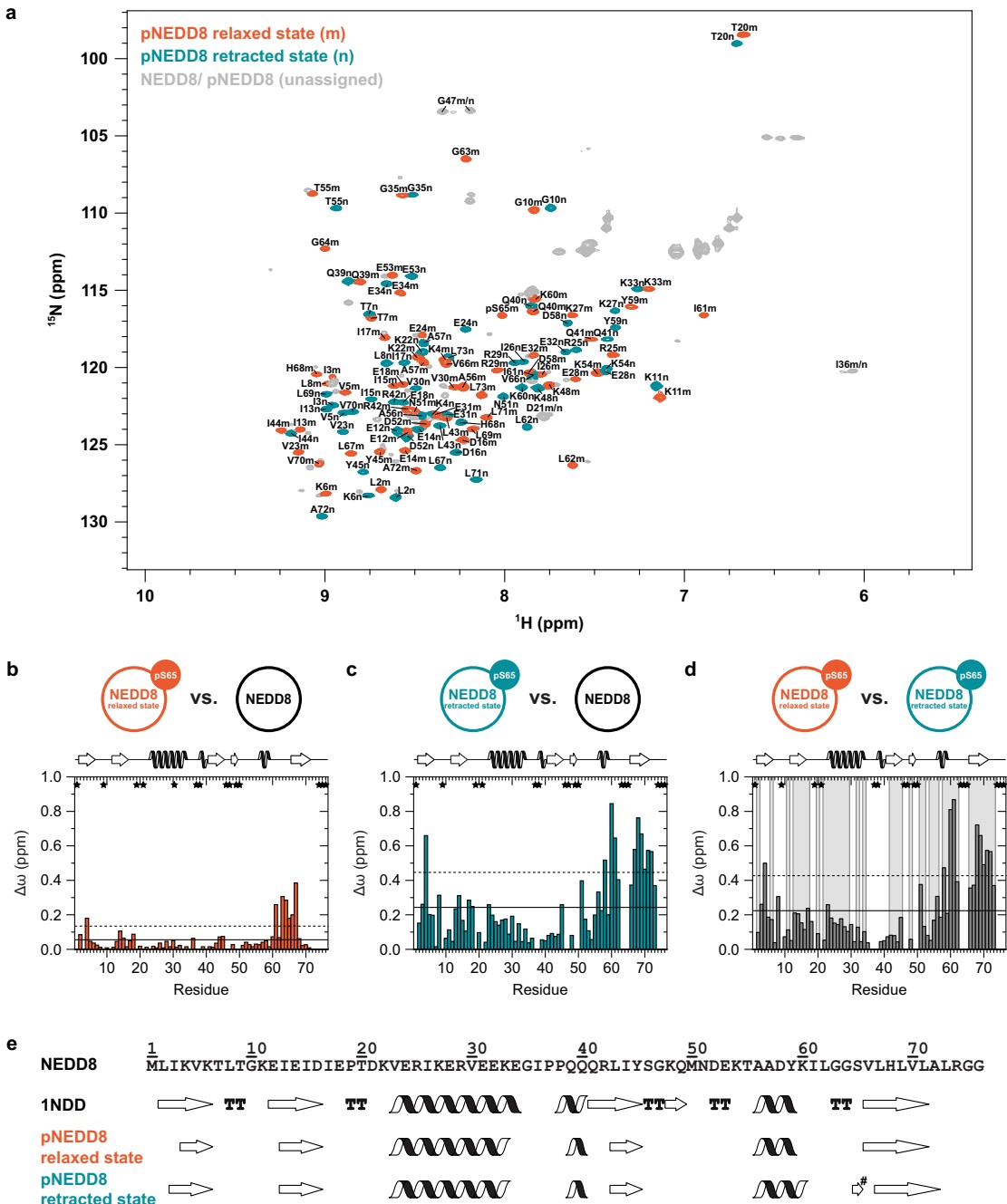

**Fig. 2 pNEDD8 adopts two conformations. a** Two-dimensional heteronuclear $^1$H–$^{15}$N HSQC spectrum of pNEDD8 showing the relaxed state (m) colored in orange and the retracted state (n) colored in green with labeling of assigned resonance signals by using the one-letter code for amino acids. Unassigned resonance signals comprising both NEDD8 and pNEDD8 are colored in grey (cf. Supplementary Fig. 3 for details). **b–d** Weighted chemical shift perturbation (CSP, Δω) mapping of pNEDD8 relaxed state vs. NEDD8 (**b**), pNEDD8 retracted state vs. NEDD8 (**c**), and pNEDD8 relaxed state vs. pNEDD8 retracted state (**d**). Δω values exceeding the horizontal straight line are larger than the mean and Δω values exceeding the horizontal dashed line are larger than the mean plus one standard deviation. Residues lacking a CSP value are indicated by an asterisk and residues revealing exchange cross-peaks in the ZZ exchange experiment (cf. Supplementary Fig. 4) are highlighted by a background bar colored in grey. Secondary structure elements according to the NEDD8 crystal structure (PDB ID 1NDD)[73] are depicted on top of the graphs. **e** Secondary structure predictions as calculated with TALOS-N are illustrated for the relaxed state and retracted state of pNEDD8. In addition, the amino acid sequence of NEDD8 and information regarding secondary structure elements on basis of the crystal structure are shown. The hashtag is highlighting secondary structure predictions that due to lack of chemical shifts, are only sequence-based.

NEDD8 upon S65 phosphorylation (Fig. 2, Supplementary Fig. 5). Moreover, considering that the proportion of the retracted state is comparable between pNEDD8 and pUb, we conclude that our structural and dynamic observations are in agreement with the ability of pNEDD8 to activate Parkin.

**Identification of the pUb and pNEDD8 interactome.** Like Ub, NEDD8 has been linked to various cellular stress pathways[30–34]. Since S65 phosphorylation affects the structural properties of Ub and NEDD8 and thereby their protein-protein interaction properties as shown for Parkin, we hypothesized that phosphorylation of

Ub and NEDD8 is more generally involved in their roles in the cellular stress response. To provide evidence for this hypothesis, we next set out to determine the interactome of pNEDD8 and pUb in comparison to their unmodified forms by an affinity enrichment strategy employing whole-cell extracts. Since cell extracts potentially contain many phosphatases, we reasoned that in addition to pNEDD8 and pUb, non-hydrolyzable phosphoserine substitute variants should be employed to avoid that during incubation a mixture of pNEDD8/pUb and NEDD8/Ub is generated. A frequently used approach is to replace respective serine residues by the phosphomimetic amino acids aspartate and glutamate. However, in the case of Ub, replacement of S65 by these amino acids does not resemble all effects of S65 phosphorylation[11], especially the characteristic retracted form of pUb could not be observed[19–21]. An obvious explanation for this shortcoming of aspartate and glutamate is that their carboxylate group mimics the single deprotonated state of the phosphate group, while adoption of the retracted form is favored by the double deprotonated state[20,21]. Since we wanted to have a phosphorylation mimic that is resistant to phosphatases but at the same time, displays the features of naturally occurring phosphoserine as closely as possible, we used again the method of genetic code expansion to incorporate a non-hydrolyzable analog of phosphoserine[26] (L-2-amino-4-phosphonobutyric acid, nhpSer; Fig. 3a) into Ub/NEDD8 at position 65. Upon expression in bacteria, quantitative incorporation of nhpSer into Ub/NEDD8 (nhpUb/nhpNEDD8) and its resistance against dephosphorylation was confirmed by ESI-MS and incubation of nhpUb/nhpNEDD8 with a promiscuous phosphatase (CIP) followed by Phos-tag gel analysis, respectively (Supplementary Fig. 6).

To gain structural information about nhpUb, we generated $^{15}$N isotopically labeled nhpUb and performed high-resolution NMR spectroscopy. The resulting $^1$H-$^{15}$N HSQC spectrum comprises a significantly higher number of cross-peaks in the backbone amide region than unmodified Ub (Fig. 3b). Importantly, most resonances of the $^1$H-$^{15}$N HSQC NMR spectrum of nhpUb could be assigned by simple superimposition with the spectrum of pUb. We therefore conclude that nhpUb also adopts two conformational states in solution and that these are in the slow exchange regime just like native pUb. Thus, from a structural point of view nhpUb is a better mimic of pUb than the phosphomimetic S65D and S65E Ub mutants[19–21]. Note that relative signal heights indicate that the two conformations are not populated in a 1:1 ratio at pH 7.4 as shown above for pUb. Again, this is likely explained by the notion that the retracted state conformation is favored by the double deprotonated state of the phosphate group[20,21]. Since nhpUb bears a phosphonate, which has a higher pKa value than the phosphate group, less molecules are present in the double deprotonated form at physiological pH and, thus, in the retracted state (12 ± 2%). Interestingly, however, the CSP patterns indicate that the retracted state conformations of nhpUb and pUb, which go along with the β5-strand slippage characteristic for S65 phosphorylation, seem to be even more similar than their corresponding relaxed state conformations (Fig. 3c, d). In line with the structural results, nhpUb bound to Parkin (Fig. 1f) and activated Parkin autoubiquitylation in vitro (Supplementary Fig. 6).

The above results indicate that nhpUb is a good structural and functional surrogate of pUb and in contrast to the natural phosphate group, the phosphonate group is refractory towards the action of phosphatases. Furthermore, because of the structural similarities between pNEDD8 and pUb, it can be safely assumed that nhpNEDD8 behaves similar to nhpUb, which is supported by the observations that it binds to Parkin (Fig. 1f) and activates Parkin autoubiquitylation (Supplementary Fig. 6). Thus, to identify interactors of pUb/pNEDD8 by AE-MS, we employed

pUb and pNEDD8 as well as the non-hydrolyzable phosphomimetic variants nhpUb and nhpNEDD8 (Supplementary Fig. 7). Unmodified Ub and NEDD8 were used for comparison and empty beads as control. In addition, all bait molecules contained an N-terminal Strep-tag for immobilization. In brief, the six bait molecules were incubated with whole-cell extracts derived from HEK293T cells, one of the most commonly used cell lines for such purposes, or SHSY5Y cells, a widely used neuronal cell model system. Bound proteins were identified by label-free quantitative MS and significantly enriched proteins by ANOVA statistics (Fig. 4a). Hierarchical clustering of the 202 and 106 significantly enriched interactors identified with HEK293T and SHSY5Y cell extracts, respectively, revealed proteins that interact preferentially with the phosphorylated (HEK293T: Cluster 1, 2, 5; SHSY5Y: Cluster 1), the unmodified (HEK293T: Cluster 4, 6, 7, 9; SHSY5Y: Cluster 3) or both forms (HEK293T: Cluster 3, 8; SHSY5Y: Cluster 2, 4) of NEDD8 or Ub (Fig. 4b).

Overall, 88 of the 106 proteins identified with SHSY5Y cell extracts were also found with HEK293T cell extracts (Fig. 4d). Moreover, the correlation of significantly enriched proteins between the two cell lines is high for the respective bait molecules indicating the robustness of our AE-MS approach. Remarkably, there is limited overlap between Ub-interacting proteins and NEDD8-interacting proteins regardless of the phosphorylation status (Fig. 4c, Supplementary Fig. 7). In fact, only when we compare proteins with an enrichment factor greater than zero for one of the Ub bait molecules versus the NEDD8 bait molecules, we find 58 overlapping proteins, while 126 are exclusive for Ub and its variants and 36 are exclusive for NEDD8 and its variants (Fig. 4e). In the pool of overlapping interactors, we find for example cullin E3 ligases. In the pool of the exclusive binders, the respective enzymes of the conjugation/deconjugation systems are located but also proteins that are not related to these processes. Together with the pairwise correlation analysis (Fig. 4c, Supplementary Fig. 7), the latter clearly indicates that despite their structural and sequence similarity, Ub and NEDD8 play different cellular roles. Furthermore, the interactomes of the pUb variants (pS65, nhpS65) are highly similar to each other, whereas unmodified Ub shows an overlapping but distinct interaction pattern, underscoring the previously described different functions of Ub and pUb[10–15]. In contrast, NEDD8 and pNEDD8 show a higher correlation in their interactomes than the pNEDD8 variants (pSer, nhpSer) do (Fig. 4c). Possible explanations for this at first glance surprising finding are partial dephosphorylation of pNEDD8 during incubation with cell extracts and/or different interactions of pNEDD8 and nhpNEDD8 due to distinct ratios of the relaxed and retracted conformations (see Discussion).

To confirm the results obtained by the AE-MS approach, we validated some of the interactions by Western blot analysis. The candidates tested represent different clusters of binding specificity, and the results obtained were in almost perfect agreement with the proteomic profiling dataset (Fig. 4f). Importantly, the data for the phospho-selective interactors HDAC6, USP16, and HSPA8 could be confirmed underlining the reliability of our AE-MS approach.

Gene ontology (GO) term enrichment analysis of the interactomes revealed that components of the chaperonin-containing T-complex are enriched with pNEDD8 and nhpNEDD8, whereas HSP70 family proteins were preferentially found for nhpNEDD8. For NEDD8 and pNEDD8, components of the COP9 signalosome complex, which is known for the deneddylation of cullins[35], were enriched, while proteins of the neddylation machinery interact with all forms of NEDD8. Proteasome and proteasome-associated proteins interact strongly with unmodified Ub but not with pUb,

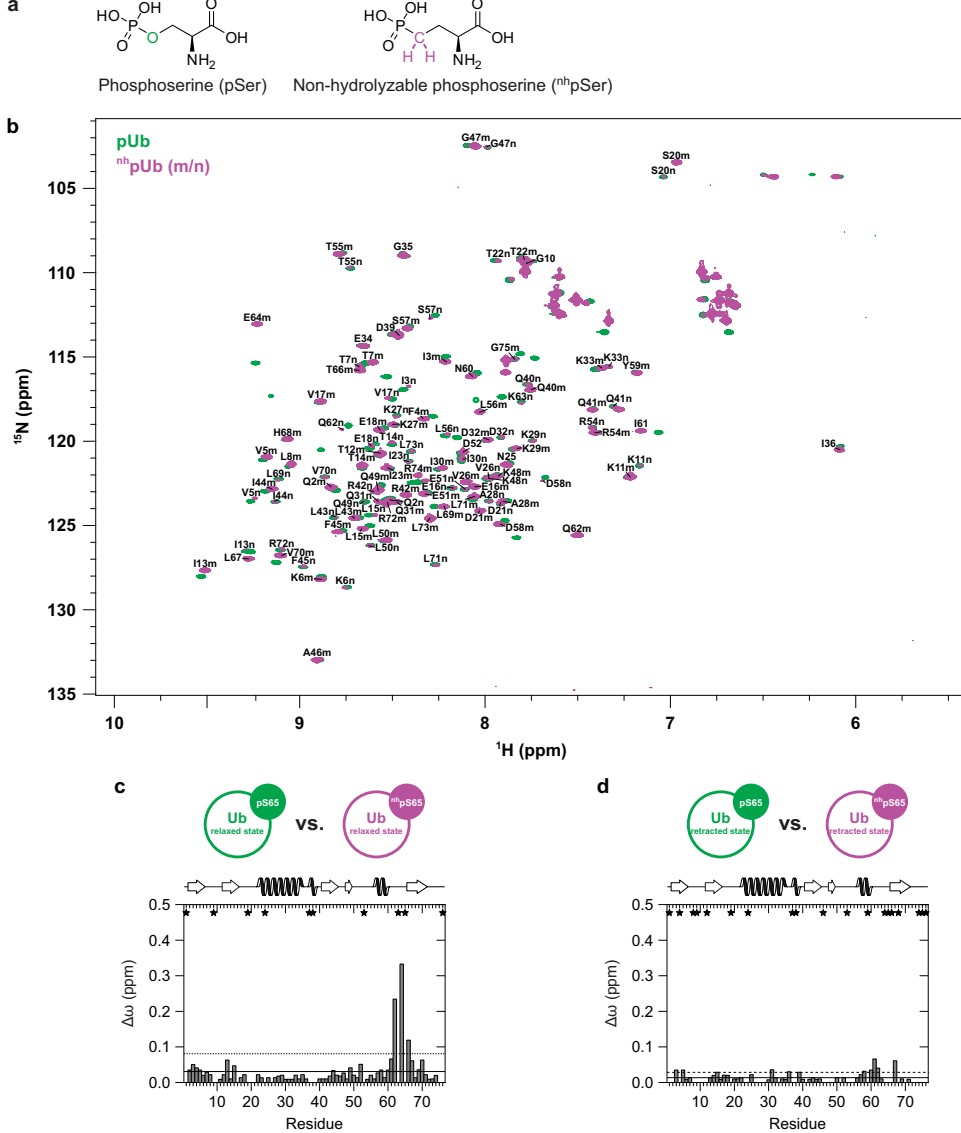

**Fig. 3 NMR spectroscopic characterization of <sup>nh</sup>pUb. a** Structure of L-O-phosphoserine (pSer) and its non-hydrolyzable analog L-2-Amino-4-phosphonobutyric acid (<sup>nh</sup>pSer). **b** Two-dimensional heteronuclear $^1$H–$^{15}$N HSQC NMR spectra of pUb (colored in green) and <sup>nh</sup>pUb (colored in purple) are superimposed. Assigned backbone amide resonance signals from the relaxed state (m) and the retracted state (n) of <sup>nh</sup>pUb are labeled in the spectrum by using the one letter code for amino acids. **c**, **d** Weighted chemical shift perturbation (CSP, Δω) mappings comparing either the relaxed states (**c**) or the retracted states of pUb and <sup>nh</sup>pUb with each other (**d**). Δω values exceeding the horizontal straight line are larger than the mean and Δω values exceeding the horizontal dashed line are larger than the mean plus one standard deviation. Residues that could not be assigned in both Ub species are indicated by an asterisk. Secondary structure elements according to the NMR solution structure of Ub (PDB ID 1D3Z)[74] are depicted on top of the graphs.

while components of the ubiquitylation machinery were found to be enriched with all forms of Ub (Fig. 5a).

**Ub/NEDD8 phosphorylation and the interaction with enzymes of the respective conjugating/deconjugating systems**. We identified 24 DUBs with some of these (15) displaying a binding preference for either pUb or unmodified Ub. Nine of the DUBs (Fig. 5b) were characterized in a previous study for their activity on isomerically phosphorylated Ub dimers[36]. Strikingly, the activity of eight of these towards the S65 phosphorylated Ub dimers correlates with the binding preference observed in our study indicating the validity of our AE-MS approach. For example, in our study, USP16 was highly enriched with pUb, while it was previously reported to have an increased activity

towards the majority of S65 phosphorylated Ub dimers[36]. In contrast, the activity of USP9x towards Ub dimers was found to be strongly inhibited by S65 phosphorylation, which perfectly fits to the enrichment of USP9x with unmodified Ub. We also identified DUBs with enhanced binding to either unmodified Ub or pUb that were not known to have a preference for either of these. USP3 and UFD1L were enriched with pUb, whereas USP14, UCHL5, USP13, USP24, PSMD7, and PSMD14 showed a binding preference for unmodified Ub. For the deneddylating enzymes identified, COPS5, COPS6, OTUB2, and SENP8 are preferentially bound to unmodified NEDD8 (Fig. 5b).

Similar to DUBs, we identified a number of E2 Ub-conjugating enzymes and E3 Ub ligases. We found in total (i.e. sum of both cell lines) 17 Ub E2 enzymes, of which UBE2R2 and UBE2Q1 were significantly enriched with pUb (Fig. 5c). Similarly, we

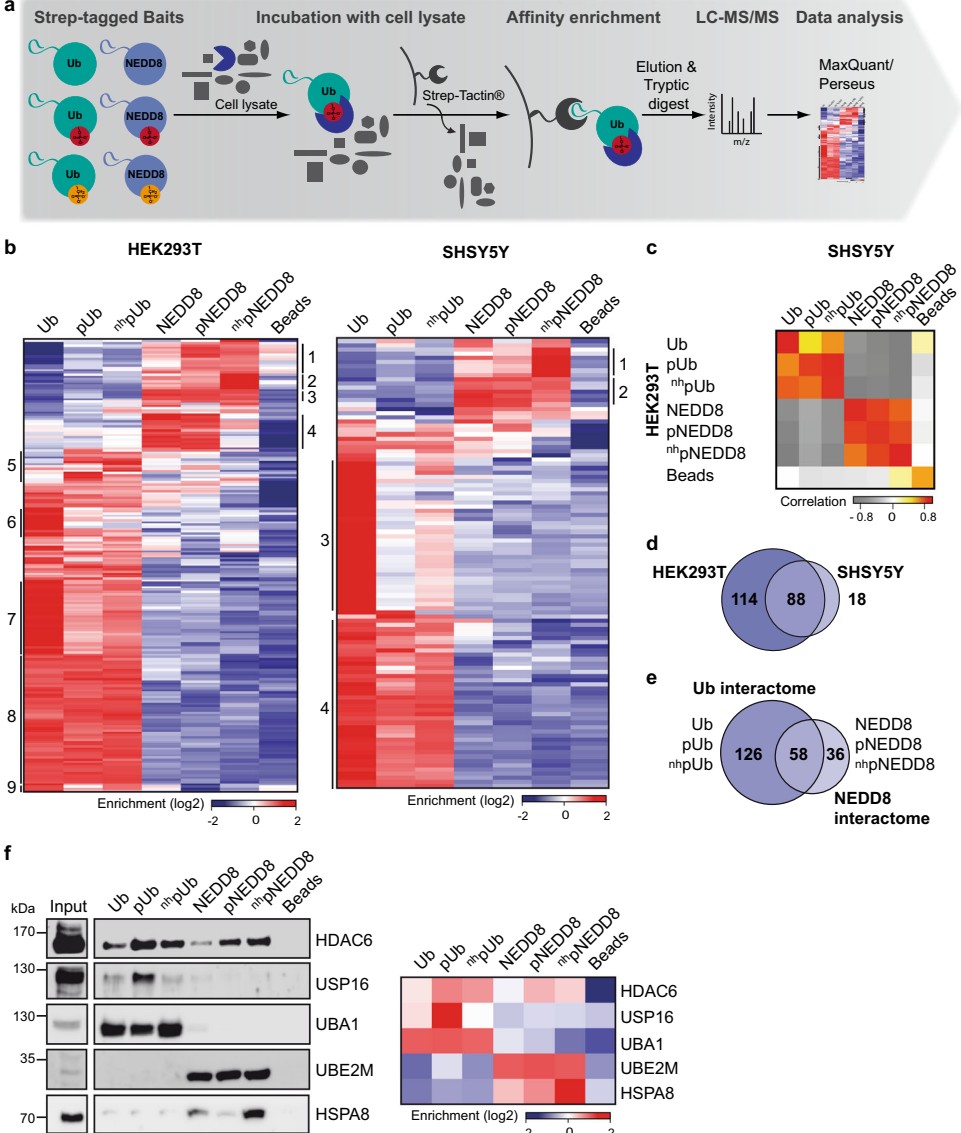

**Fig. 4 Identification of the interactome of phosphorylated Ub and NEDD8. a** Schematic overview of the AE-MS workflow for the identification of pUb and pNEDD8 interacting proteins. The bait molecules Ub, pUb, nhpUb, NEDD8, pNEDD8, or nhpNEDD8 were used as affinity matrices. Empty beads were used as control. $n=3$ biologically independent experiments measured in technical duplicates. **b** Hierarchical clustering of statistically significant interacting proteins (rows) and the different bait molecules (columns) in HEK293T and SHSY5Y cell lysates. Thresholds for the ANOVA statistics were set as follows: FDR = 0.02, S0 = 1. Enrichment is indicated in red, whereas lack of enrichment is indicated in blue. Numbers of the different clusters are indicated on both sides. **c** Plot of the heatmap of pairwise correlation between significantly enriched interactors for each bait in HEK293T and SHSY5Y cell extracts. A strong correlation is indicated in red, medium correlation in yellow, no correlation in white and anti-correlation in grey. **d** Venn diagram showing the overlap of significantly enriched interactors for the pull-down performed with HEK293T and SHSY5Y cell lysates. **e** Overlap between the combined significantly enriched interactors of all Ub and all NEDD8 variants. HEK293T and SHSY5Y data were combined. For the heatmap of pairwise correlation between significantly enriched interactors for each bait see Supplementary Fig. 7. **f** Confirmation of interactions by Western blot analysis. 10% of the elution fraction of the affinity enrichment assays were subjected to Western blot analysis with antibodies specific for the respective protein. Input represents 1% of the HEK293T cell lysate used for affinity enrichment. The Western blot shown is representative of three independent experiments. Source data are provided as a Source Data file.

found UBE2M, one of the two known NEDD8 E2 enzymes, enriched with all NEDD8 variants. The 47 E3 Ub and/or NEDD8 ligases found (Fig. 5d) represent proteins binding to NEDD8/pNEDD8 (HEK293T Cluster 4 (3); SHSY5Y Cluster 2 (2)), unmodified Ub (HEK293T Cluster 6 (2); SHSY5Y Cluster 3 (7)), and all Ub variants (HEK293T Cluster 8 (22); SHSY5Y Cluster 4 (18)). We also identified more or less the entire 26 S proteasome (Fig. 5e). Remarkably, the proteasomal subunits were exclusively identified with unmodified Ub (Fig. 4b; HEK293T: Cluster 6/7/9, SHSY5Y: Cluster 3). This is a strong indication that pUb and

substrates attached to it are not targeted for proteasomal degradation.

**Interactors of pUb and pNEDD8**. Since the binding specificity of the DUBs identified correlated well with the literature, we had a closer look at the cluster of proteins that preferentially interact with the phosphorylated forms of NEDD8 and Ub (Fig. 6a, b). Plotting the known string interaction network[37] of these proteins (Fig. 6c), we find among them the UFD1L-NPLOC4-VCP complex, which binds to ubiquitylated substrates and extracts them

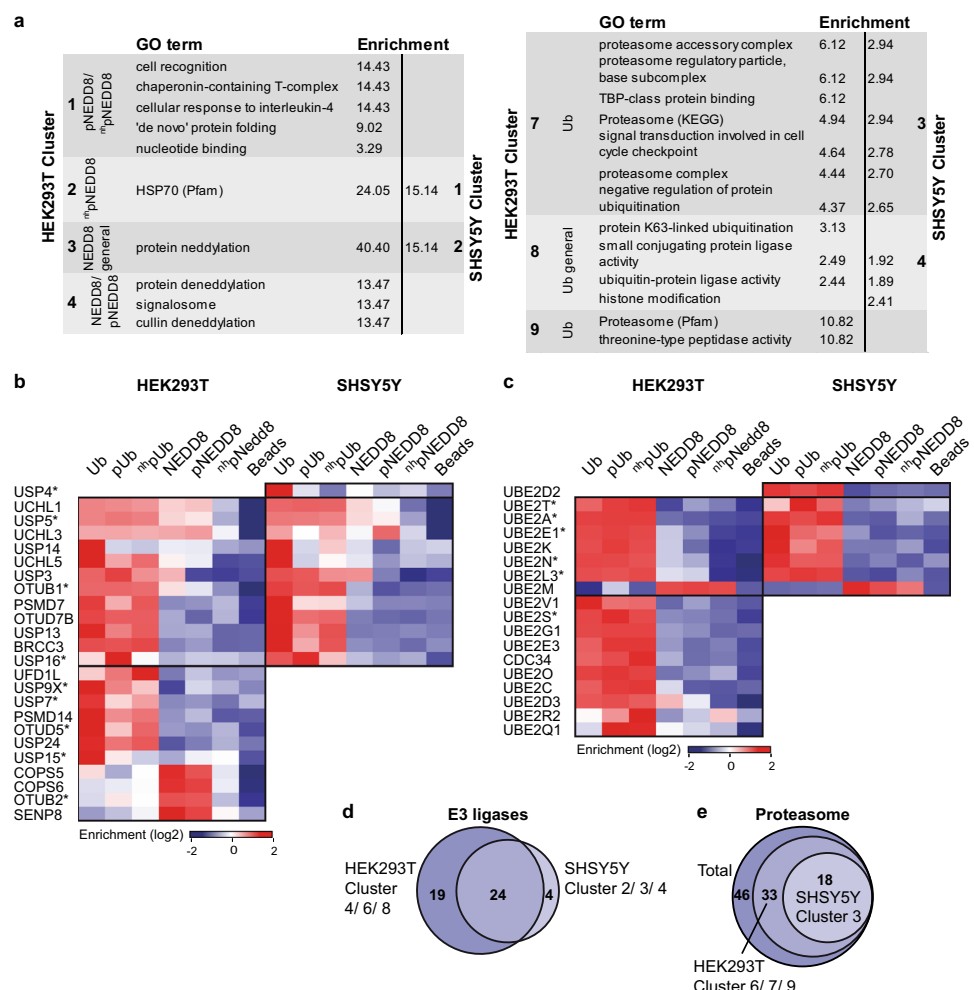

**Fig. 5 Enzymes of Ub/NEDD8 conjugation/deconjugation and their binding preferences. a** GO terms, KEGG, and Pfam annotations with enrichment in the identified clusters (Fig. 4b) including biological processes, protein complexes, and families. For HEK293T clusters 5 and 6, none of the GO terms was enriched. **b** Hierarchical clustering of DUBs identified in HEK293T and SHSY5Y AE-MS experiments. The DUBs with an asterisk were found previously to be either activated or inhibited by phosphorylation at S65[36]. **c** Hierarchical clustering of E2 enzymes identified. The E2s with an asterisk were studied before with phosphorylated Ub as a substrate[19]. Thresholds for the ANOVA statistics (**b,c**) were set as follows: FDR = 0.02, S0 = 1. Enrichment is indicated in red, whereas lack of enrichment is indicated in blue. $n = 3$ biologically independent experiments measured in technical duplicates. **d** Venn diagram overview showing all E3s found in either one of the cell lines or in both AE-MS experiments. In which clusters (Fig. 4b) the E3s are mainly represented in is indicated. **e** Venn diagram showing the amount of proteasomal subunits found in the two experiments and the clusters (Fig. 4b) they are represented in. Source data are provided as a Source Data file.

from their subcellular structures[38]. Interestingly, all three components of this complex are enriched in the nhpUb/pUb samples indicating that the complex is involved in the downstream signaling of Ub phosphorylation. Note that our data do not allow us to conclude, which of the components of the UFD1L-NPLOC4-VCP complex interact(s) with pUb, as the affinity enrichment was performed under non-denaturing conditions and, thus, also proteins that indirectly interact with nhpUb/pUb are enriched. Other proteins that we found enriched for pUb/nhpUb are FAF1, PCNA, NAP1L4, WAC, CUL4B, and AMBRA1, an interactor of Parkin and autophagy regulator[39]. We also observed enrichment of HDAC6 with both pUb and pNEDD8. HDAC6 is a critical factor for autophagic clearance of Ub-decorated aggregates/aggresomes[40] and its UBP domain shows the highest affinity for mono-Ub among all known human Ub-binding proteins[41]. Moreover, HDAC6 is also a known interactor of Parkin[42] and involved in mitophagy[43], suggesting that pUb/pNEDD8 are involved in targeting HDAC6 to mitochondria and that pUb/

pNEDD8 are not only involved in mitophagy, but more generally in the cellular response to proteotoxic stress. This is supported by a recent study indicating that NEDD8 and HDAC6 are involved in the cellular response to proteotoxic stress by forming cytosolic aggresome-like structures linked to the aggresome-autophagy flux[44].

NUB1 showed a strong enrichment with nhpNEDD8. Since NEDD8 was reported to serve as a signal for proteasomal degradation via the NUB1 pathway[45], this may suggest that pNEDD8 either stimulates NUB1-mediated degradation or interferes with it. However, the role of NUB1 in targeting neddylated proteins for degradation has been challenged recently[33], which may explain our observation that proteasomal subunits were not enriched with any of the NEDD8 variants. In that study[33], overexpression of NUB1 resulted in the repression of neddylation and the promotion of ubiquitylation and proteasomal interaction of a model substrate, especially under proteotoxic stress. Thus, one may speculate that pNEDD8 binds NUB1 under

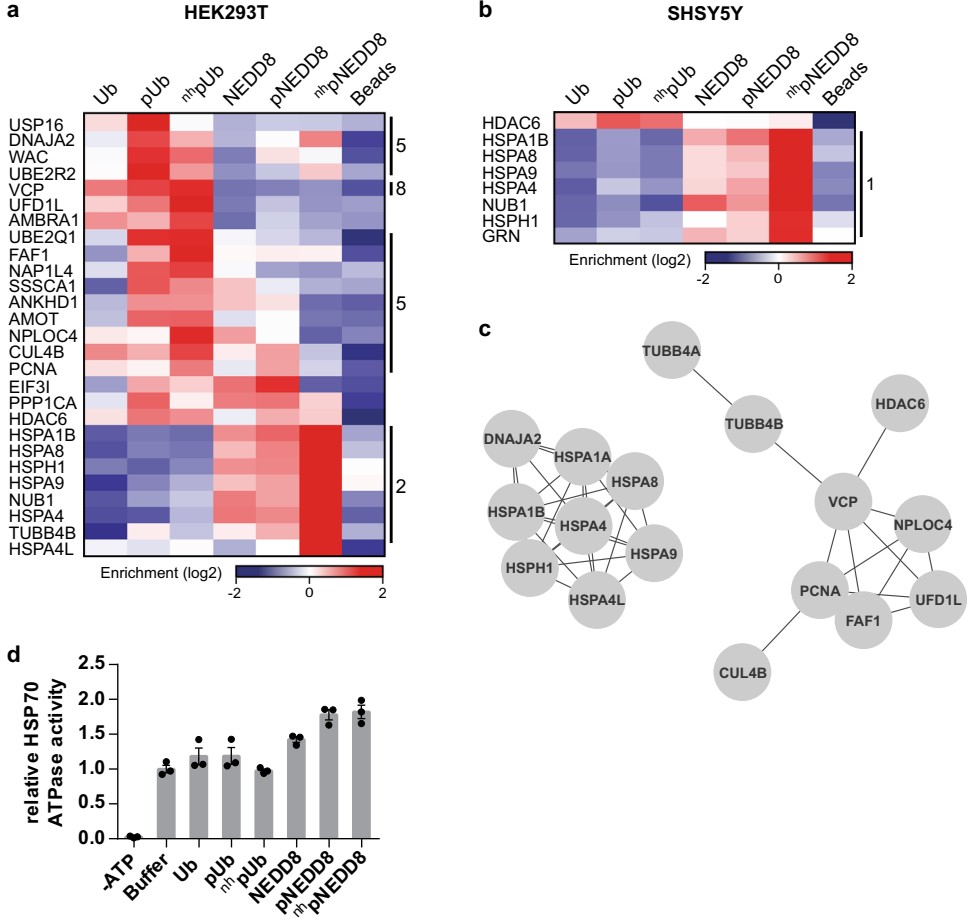

**Fig. 6 pNEDD8 but not pUb stimulates HSP70 ATPase activity. a**, **b** Selected proteins from the AE-MS experiments performed with HEK293T (**a**) or SHSY5Ycell (**b**) extracts with a binding preference for pUb or pNEDD8. Thresholds for the ANOVA statistics were set as follows: FDR = 0.02, S0 = 1. Enrichment is indicated in red, whereas lack of enrichment is indicated in blue. Clusters are as in Fig. 4b. **c** String interaction networks showing only interactions with the highest confidence (interaction score 0.9) for the proteins selected in **a** and **b**. **d** The relative stimulation of the ATPase activity of HSPA8, a member of the HSP70 family, was measured in presence of Ub, pUb, nhpUb, NEDD8, pNEDD8, and nhpNEDD8 and compared to HSP70 basal ATPase activity ($n = 3 \pm$ SEM independent experiments). Source data are provided as a Source Data file.

certain stress conditions, thereby allosterically stimulating NUB1's function in proteasomal degradation.

**pNEDD8 interacts with HSP70 family proteins and stimulates HSP70 ATPase activity.** Among the proteins enriched with nhpNEDD8 (Figs. 6a, b), HSP70 family members (HSPA1A/B, HSPA4, HSPA4L, HSPA8, HSPA9) and their co-factors (co-chaperone DNAJA, nucleotide-exchange factor HSPH1) are found, which is also visible in the network depiction of known string interactions (Fig. 6c). Interestingly, it was recently reported that NEDD8 binds to the ATPase domain of HSP70 at the same region as co-chaperones and stimulates HSP70's ATPase activity[32]. Because of the strong enrichment of HSP70 with nhpNEDD8, we hypothesized that phosphorylation of NEDD8 increases its affinity to HSP70 and its potential to stimulate HSP70's ATPase activity. Indeed, ATPase activity assays showed that pNEDD8 variants stimulate HSPA8, which was used as representative of HSP70 family members, more potently than unphosphorylated NEDD8 (Fig. 6d, Supplementary Fig. 7) and this effect depends on the I44 hydrophobic patch, which is required for NEDD8 to interact with HSP70[32] (Supplementary Fig. 7). In contrast, Ub and its phosphorylated variants showed no significant stimulation indicating the specificity of this effect for NEDD8.

**Discussion**

Phosphorylation of Ub has so far been studied mainly in the context of PINK1/Parkin-mediated mitophagy. Thereby, the principle of "modifying the modifier" was established and the complexity of the Ub code extended[46–48]. Here we show that like Ub, NEDD8 is phosphorylated by PINK1 at S65 and that pNEDD8 stimulates PINK1-mediated Parkin phosphorylation and in addition acts as an allosteric activator of Parkin. While this suggests that phosphorylation of NEDD8 contributes to the mitochondrial stress response, NEDD8 has been associated with several other stress pathways and, thus, it seems likely that NEDD8 phosphorylation plays a more general role in the cellular stress response. For instance, it was reported that upon neuronal stress, IL-1β fosters NEDD8-induced Parkin activation by inducing translocation of NEDD8 from the nucleus to the cytoplasm[31] and that NEDD8 and Parkin colocalize in the cytoplasm resulting in Parkin neddylation and activation[18]. However, neither study investigated the phosphorylation status of NEDD8. Furthermore, in silico simulations predicted that NEDD8 binds to the same region of Parkin as pUb and induces the release of Parkin's Ub-like domain[31], which is critical for subsequent phosphorylation and complete activation of Parkin. We now confirm and refine this prediction by showing that pNEDD8 but not unmodified NEDD8 activates Parkin and propose that upon neuronal stress, NEDD8 is phosphorylated which leads to activation of Parkin.

In analogy to pUb[19–21], structural analysis of pNEDD8 revealed a conformational equilibrium between a relaxed conformation that is also adopted by unmodified NEDD8 and Ub and a C-terminally retracted conformation. Since the retracted conformation can also be detected with unmodified Ub, though at a rather low abundance (~0.68% at 45 °C)[22], we propose that phosphorylation stabilizes this alternative conformation, rather than inducing it. In any case, the presence of two conformations prompted us to determine the interactome of pNEDD8 and pUb in comparison to their unmodified counterparts using whole cell lysates. Since whole cell lysates are rich in phosphatases, we reasoned that the use of non-hydrolyzable pNEDD8/Ub variants in addition to their natively phosphorylated forms may prove beneficial. Notably, the structural analysis revealed that also nhpUb exists in the C-terminally retracted conformation, though to a lower extent than pUb. Nevertheless, the incorporation of nhpSer provided a good compromise between stability against phosphatases and the structural and functional effects of natural phosphorylation. Yet, subsequent identification of the interactomes by AE-MS revealed differences between natively phosphorylated Ub/NEDD8 and the respective non-hydrolyzable phosphorylated variants. A possible explanation for these differences is that nhpUb shows a different ratio of the retracted and the relaxed form (~ 1:7) than pUb (~ 1:1). The lower amount of the C-terminally retracted form in the nhpUb sample will affect the binding of interactors that prefer one conformation to the other. Because of the structural similarity of Ub and NEDD8, we assume that the same argument holds true for pNEDD8 and nhpNEDD8. As abovementioned, another possibility is that the natively phosphorylated Ub/NEDD8 variants are partly dephosphorylated during incubation with whole-cell lysates. The resulting mixture of phosphorylated and unphosphorylated forms of Ub/NEDD8 will for obvious reasons result in a different pattern of interactors compared to the non-hydrolyzably phosphorylated Ub and NEDD8 variants. Anyway, among the interacting proteins we found DUBs with a binding preference for either phosphorylated or unphosphorylated Ub, some of which were previously assayed for their activity against S65-phosphorylated Ub dimers[36]. DUB activity observed in this previous study correlates well with the binding preference observed in our study, indicating that the increased activity of certain DUBs towards S65-phosphorylated Ub dimers is due to a higher affinity for pUb. The other pUb-specific interactors that we found are involved in various cellular processes, including different stress response pathways. Thus, it is tempting to speculate that the role of Ub phosphorylation is not restricted to mitophagy but plays a more general role in cell regulatory pathways.

Although HSP70 family proteins were strongly enriched by nhpNEDD8 but not by pNEDD8, we showed that both stimulated the ATPase activity of HSP70 more strongly than unphosphorylated NEDD8. This apparent contradiction between enrichment and HSP70 stimulation is readily explained by the notion that pNEDD8 was in part dephosphorylated during the affinity enrichment procedure, underlining the advantage of using both hydrolyzable and non-hydrolyzable variants for the identification of interacting proteins. Furthermore, we observed enrichment of the de-neddylase SENP8 (NEDP1)[49] with NEDD8, but not with nhpNEDD8. In a recent study, it was shown that NEDD8 conjugation/deconjugation plays an important role in DNA damage-induced apoptosis[32]. Furthermore, SENP8 expression is induced by DNA double-strand break-inducing agents thereby restricting NEDD8 chain formation, while free NEDD8 binds to HSP70 and stimulates its ATPase activity[32]. We therefore determined whether similar to mitochondrial stress, induction of DNA double-strand breaks affects the NEDD8 phosphorylation status in cells. Although this appears to be the case, we observed a decrease in NEDD8 phosphorylation rather than an increase under the conditions used (Supplementary Fig. 7). Although this does not exclude the possibility that pNEDD8 contributes to DNA damage-induced stimulation of HSP70 activity, it suggests that the effect of pNEDD8 on HSP70's ATPase activity is more relevant in other stress response pathways that require HSP70 proteins to maintain protein homeostasis[50].

Ub and NEDD8 have been linked to various cellular stress pathways[30–34]. While numerous studies have studied the role of covalent attachment of Ub and NEDD8 to other proteins, it was recently reported that unanchored NEDD8 chains[32] and acetylated unanchored NEDD8 trimers[30] are involved in the cellular stress response as well. Our data indicate that also phosphorylation of Ub and NEDD8 plays a more prominent role in stress signaling as so far assumed and support the notion that pUb and pNEDD8 do not need to be covalently attached to other proteins to exert their function.

## Methods

**Plasmid construction**. The cDNA encoding full-length human Parkin was inserted into pGEX-2TK. The codon-optimized cDNA of human UCHL3 and the cDNA encoding the ATPase domain of the HSP70 family member HSPA8 (residue 1–402, including the ATPase activity stimulating linker domain[51]) were inserted into pET15b.

Codon-optimized cDNA of SepRS[26] and a gene block encoding tRNA$^{Sep}_{CUA}$[26] were cloned into the pEVOL vector backbone[52] to yield pEVOL Sep. A codon-optimized cDNA of EF-Sep containing a tac promoter and an rrnB terminator was cloned into the pRSFDuet vector to yield pRSF EF-Sep. Additionally, a codon-optimized cDNA of serB was cloned into pRSF EF-Sep to give pRSF EF-Sep serB. A codon-optimized cDNA of Ub with a C-terminal His$_6$-tag was inserted into the pGEX-2TK backbone destroying the GST cDNA to yield pKS Ub-His$_6$. The cDNA of human NEDD8 (1-76) was inserted into pKS to yield pKS NEDD8-His$_6$. Replacement of the S65 codon by TAG was performed by site-directed mutagenesis to yield pKS Ub S65TAG-His$_6$ and pKS NEDD8 S65TAG-His$_6$. For the NEDD8 LIA and pNEDD8 LIA variants, the L8 and I44 codons were replaced by codons for A by site-directed mutagenesis to yield pKS NEDD8 LIA-His$_6$ and pKS NEDD8 LIA S65TAG-His$_6$.

The N-terminal Strep-tag II was inserted in front of the Ub/NEDD8 variants by site-directed mutagenesis (Q5 Site-Directed Mutagenesis Kit, New England Biolabs).

All oligonucleotides used for cloning are listed in Supplementary Table 1.

**Bacterial expression and protein purification**. Human His$_6$-tagged UBA1 and C-terminally His$_6$-tagged UbcH7 were expressed in E. coli BL21 (DE3) RIL. Both were purified via immobilized metal ion chromatography (IMAC)[53,54]. Ub was expressed in E. coli BL21 (DE3) and purified via IMAC followed by cation exchange chromatography[55].

N-terminal His$_6$-tagged UCHL3 was expressed in E. coli BL21 (DE3) for 5 h at 37 °C. Cells were harvested, resuspended in lysis buffer (25 mM Tris-HCl pH 7.5, 300 mM NaCl, 10 mM imidazole, 1% (v/v) Triton-X100) and lysed by sonication. The lysate was cleared by centrifugation and the His$_6$-tagged UCHL3 was purified via a 5 ml HisTrap FFcrude (GE Healthcare) with a linear gradient from 10 to 500 mM imidazole in 25 mM Tris-HCl pH 7.5, 300 mM NaCl. UCHL3-containing fractions were pooled, dialyzed against 25 mM Tris-HCl pH 8, 300 mM NaCl, 1 mM DTT, and stored at −80 °C until further use.

NEDD8/NEDD8 LIA were expressed in E. coli BL21 (DE3) for 6 h at 37 °C. Cells were harvested, resuspended in lysis buffer (50 mM Tris-HCl pH 7.4, 150 mM NaCl, 30 mM imidazole, 1 mg/ml Aprotinin/Leupeptin, 1 mg/ml Pefabloc, 1% (v/v) Triton X-100) and lysed by sonication. The lysate was cleared by centrifugation (39,000× g, 4 °C, 30 min) and the C-terminal His$_6$-tagged protein was purified via a 5 ml HisTrap HP with a step elution from 30 mM to 242 mM imidazole in 50 mM Tris-HCl pH 7.4, 150 mM NaCl. NEDD8-containing fractions were pooled, UCHL3 was added to a final concentration of 75 μg/ml to remove the C-terminal His-tag and the samples were incubated for 1–4 h at room temperature. After cleavage, the samples were dialyzed overnight at 4 °C against 25 mM Tris-HCl pH 7.5, 300 mM NaCl. NEDD8 was further purified by size-exclusion chromatography (HiLoad 16_600 Superdex 75 pg). The purity of the fractions was confirmed by SDS-PAGE; fractions were pooled and concentrated using Amicon Ultra (3 kDa MWCO). The concentration was determined with the Pierce™ BCA Protein Assay Kit (Thermo Fischer) and commercial Ub (Sigma) as standard.

Pediculus humanus corporis (Ph)PINK1 (115-575) was a gift from David Komander (Addgene plasmid # 110750; http://n2t.net/addgene:110750; RRID:Addgene_110750). GST-tagged PhPINK1[56] and GST-tagged Parkin were

expressed in *E. coli* BL21 (DE3) RIL. Cells were grown at 30 °C in 2xYT medium until an $OD_{600} = 0.7$ was reached. After harvesting the cells, resuspending them in 42 °C warm 2xYT medium and incubating them for 30 min at 42 °C, the culture was diluted to an $OD_{600} = 0.3$ with ice-cold 2xYT medium. Upon incubation, at 16 °C until $OD_{600} = 0.6$ was reached, protein expression was induced by the addition of 10 µM IPTG, 200 µM ZnCl$_2$. The culture was harvested after 48 h expression at 16 °C. The GST-tagged proteins were purified via Glutathione Sepharose 4B beads (GE Healthcare)[10,56].

N-terminal His$_6$-tagged HSP70 ATPase domain (residue 1-402, including the ATPase activity stimulating linker domain[51]) was expressed in *E. coli* BL21 RIL (DE3) overnight at 20 °C. Cells were harvested, resuspended in lysis buffer (50 mM Tris-HCl pH 8, 300 mM NaCl, 6 mM MgCl$_2$, 20 mM imidazole, 1 mM DTT, 10% (v/v) glycerol, 1 mg/ml Aprotinin/Leupeptin, 1 mg/ml Pefabloc) and lysed by sonication. The lysate was cleared by centrifugation and the His$_6$-tagged HSP70 ATPase domain was purified via a 5 ml HisTrap HP (GE Healthcare) with a linear gradient from 20–500 mM imidazole in 50 mM Tris-HCl pH 7.5, 300 mM NaCl, 6 mM MgCl$_2$, 1 mM DTT. HSP70 ATPase domain-containing fractions were pooled, dialyzed against 100 mM Tris-HCl pH 7.5, 20 mM KCl, 6 mM MgCl$_2$, 0.5 mM DTT, and stored at −20 °C until further use.

**Generation of phosphorylated and non-hydrolyzable phosphorylated Ub and NEDD8.** The site-specific incorporation of L-O-Phosphoserine (pSer) and its non-hydrolyzable analog DL-AP4 ($^{nh}$Sep) at position 65 of Ub and NEDD8 was achieved by using a SepRS/ tRNA$^{Sep}_{CUA}$ pair for amber codon suppression.

*E. coli* B834 (DE3) containing the corresponding pKS vector, pRSF EF-Sep and pEVOL Sep were grown overnight (37 °C, LB + Carb/Kan/Cm: LB medium containing 100 µg/L carbenicillin/ 34 µg/L kanamycin/ 34 µg/L chloramphenicol). The expression culture was inoculated to an $OD_{600} = 0.1$ in LB + Carb/Kan/Cm and incubated at 37 °C until $OD_{600} = 0.4$ was reached. Arabinose and Sep were added to a final concentration of 0.02% and 2 mM, respectively, and the culture was incubated for 30 min at 37 °C. Then, IPTG and Sep were added to reach a final concentration of 1 mM and 4 mM, respectively. After further incubation (37 °C, 6 h), cells were harvested by centrifugation, resuspended in lysis buffer (50 mM Tris-HCl pH 7.4, 150 mM NaCl, 30 mM imidazole, 1 mg/ml Aprotinin/ Leupeptin, 1 mg/ml Pefabloc, 1% (v/v) Triton X-100) and lysed by sonication. The lysate was cleared by centrifugation (39,000× g, 4 °C, 30 min) and pUb-His/ pNEDD8-His was purified using a 5 ml HisTrap HP (Äkta pure25) with a step from 30–242 mM imidazole in 50 mM Tris-HCl pH 7.4, 150 mM NaCl for elution. Protein-containing fractions were pooled, UCHL3 was added to a final concentration of 75 µg/ml to remove the C-terminal His-tag, and the samples were incubated for 1–4 h at room temperature. After cleavage, pUb samples were dialyzed overnight at 4 °C against 20 mM Tris-HCl pH 8.7, and pNEDD8 samples were dialyzed against 25 mM Tris-HCl pH 7.5, 300 mM NaCl. Note that due to dephosphorylation events during the expression and purification procedure, pUb/pNEDD8 samples contain traces of unmodified Ub/NEDD8. Therefore, pUb samples were further purified via anion exchange chromatography (AEX; MonoQ 5_50 gl) with a linear gradient from 20 mM Tris-HCl pH 8.7 to 25 mM Tris-HCl pH 7.5, 50 mM NaCl, which allows for the separation of pUb from unmodified Ub, UCHL3, and the cleaved His$_6$-tag. Since in contrast to Ub, NEDD8 has a tendency to aggregate under the conditions of AEX, pNEDD8 samples were subjected to size exclusion chromatography (SEC) using a Superdex S75 10_300 gl. While this allows the separation of pNEDD8 from UCHL3 and the cleaved His$_6$-tag, it does not separate pNEDD8 and NEDD8. Thus, pNEDD8 samples contained traces of unmodified NEDD8. pUb/pNEDD8 fractions were confirmed by Phos-tag™ SDS-PAGE, pooled, and concentrated using Amicon Ultra (3 kDa MWCO). The concentration was determined with the BCA Protein Assay Kit (Thermo Fisher) and commercial Ub as standard. The correct masses and site of incorporation of Sep were confirmed by ESI-MS and LC-MS/MS.

The incorporation of the non-hydrolyzable analog L-AP4 ($^{nh}$pSer) was performed as described above, except for using pRSF EF-Sep serB instead of pRSF EF-Sep. DL-AP4 was synthesized according to literature[57] and used at a final concentration of 8 mM. Note that *E. coli* contains low levels of endogenous phosphoserine which competes with $^{nh}$pSer for incorporation[26]. AEX chromatography allowed the separation of $^{nh}$pUb from pUb and unmodified Ub, whereas $^{nh}$pNEDD8, pNEDD8, and unmodified NEDD8 could not be separated via SEC (see above). Thus, $^{nh}$pNEDD8 samples contained traces of pNEDD8 and unmodified NEDD8.

All variants were expressed and purified with and without an N-terminal Strep-tag II. For the C-terminally His-tagged proteins, the UCHL3 cleavage step was skipped.

**Phos-tag™ SDS-PAGE.** 1 µg of phosphorylated protein was incubated without or with 5U alkaline phosphatase (New England Biolabs) for 60 min, then the reaction was quenched with 6x reducing SDS loading buffer (50 mM Tris-HCl pH 6.8, 10% (v/v) glycerol, 2 % (w/v) SDS, 1 % (v/v) β-mercaptoethanol, 0.05 % (w/v) brom-phenol blue) and samples were analyzed by electrophoresis on a 15% poly-acrylamide gel containing 25 µM Phos-tag acrylamide (Wako Chemicals) and 50 µM MnCl$_2$.

**In vitro Parkin activity assay.** For in vitro ubiquitylation, 100 nM UBA1, 3 µM UbcH7, 318 nM GST-Parkin, GST-Parkin ΔUbl or GST-Parkin H302A were incubated in 25 mM Tris-HCl pH 7.5, 50 mM NaCl, 0.6 mM DTT, 2 mM ATP and 2 mM MgCl$_2$. The total amount of Ub and NEDD8 variants was kept at 10 µg (39 µM) with the mass percentages of pUb and pNEDD8 indicated in the respective figures. The reaction was started by the addition of Ub in a total reaction volume of 30 µl and incubated for 60 min at 30 °C. Reactions were stopped by the addition of 6× reducing SDS loading buffer (50 mM Tris-HCl pH 6.8, 10% (v/v) glycerol, 2% (w/v) SDS, 1% (v/v) β-mercaptoethanol, 0.05% (w/v) bromphenol blue). 2/3 of the mixture were electrophoresed in 12.5 or 15% SDS-PA gels and stained with Coo-massie blue. 1/3 of the mixture was electrophoresed in the same manner but sub-sequently subjected to Western blot analysis using the monoclonal anti-Ub antibody FK2 (1:1,000; #ST1200, Merck) and the secondary antibody anti-mouse HRP (1:15,000; #115-035-062, Jackson ImmunoResearch). Amersham Imager 600 Ana-lysis Software (GE) was used for quantification.

**In vitro phosphorylation by *Ph*PINK1.** For in vitro phosphorylation of Ub/ NEDD8, 140 µM GST-*Ph*PINK1, 10 mM ATP, 10 mM MgCl$_2$, and 2 µg of either Ub or NEDD8 were incubated at 30 °C in 50 mM Tris-HCl pH 7.5, 50 mM NaCl, 1 mM DTT for the indicated times. Reactions were stopped by the addition of 6x SDS loading buffer and analyzed via Phos-tag PAGE. For in vitro phosphorylation of GST-Parkin, 0.25 µM GST-PhPINK1, 10 mM ATP, 10 mM MgCl$_2$, and 2.5 µM GST-Parkin were incubated at 37 °C in 50 mM Tris-HCl pH 8.7, 200 mM NaCl, 5 mM DTT for the indicated times in absence or presence of either pUb, pNEDD8 or NEDD8. Reactions were stopped by the addition of 6x SDS loading buffer and analyzed via Phos-tag PAGE. Amersham Imager 600 Analysis Software (GE) was used for quantification.

**Parkin binding assay.** For affinity enrichment of Parkin, 5 µM (2.5 µg) of the respective Strep-Ub / NEDD8 variant was incubated with 10 µM Parkin (wild-type, ΔUbl or H302A) for 30 min on ice. The mixture was added to 10 µl Strep-Tactin XT 4Flow beads (iba), preequilibrated with two times 200 µl binding buffer (1x PBS, 2 mM MgCl$_2$, 1 mM DTT). Upon incubation at RT for 3 h in an overhead shaker, the beads were washed three times with 200 µl lysis buffer each. Bound proteins were eluted by incubating the beads for 5 min on ice with two times 30 µl elution buffer (100 mM Tris-HCl pH 8.0, 150 mM NaCl, 1 mM EDTA, 2.5 mM desthiobiotin). The eluates were combined and 25% of them were applied to SDS-PAGE followed by Western blot analysis using an antibody specific for Parkin (anti-Parkin PRK8, 1:1,000; (#sc-32282, Santa Cruz)) and as secondary antibody anti-mouse HRP (1:15,000; #115-035-062, Jackson ImmunoResearch).

**Isotopically labeled proteins for NMR studies.** For preparation of $^{13}$C/ $^{15}$N-labeled pNEDD8, *E. coli* strain harboring the respective expression construct was grown in LB medium at 37 °C until $OD_{600} = 1$. This culture was used to inoculate $^{13}$C/$^{15}$N M9 Medium (33.7 mM Na$_2$HPO$_4$, 22 mM KH$_2$PO$_4$, 8.55 mM NaCl, 9.35 mM $^{15}$NH$_4$Cl, 1 mM MgSO$_4$, 0.3 mM CaCl$_2$, 134 µM EDTA, 31 µM FeCl$_3$, 6.2 µM ZnCl$_2$, 0.76 µM CuCl$_2$, 0.42 µM CoCl$_2$, 1.62 µM H$_3$BO$_3$, 0.081 µM MnCl$_2$, 1 mg/L biotin, 1 mg/L thiamine, 0.2% (w/v) $^{13}$C glucose) and was grown overnight at 37 °C. 100 ml of the culture were diluted with 900 ml pre-warmed $^{13}$C/$^{15}$N M9 Medium. Cells were grown at 37 °C until $OD_{600} = 0.4$ was reached. Then, 0.02% arabinose and 2 mM Sep were added, and cells were incubated for further 30 min at 37 °C. Protein expression was induced by the addition of 1 mM IPTG and 4 mM Sep (final concentrations). Cells were harvested after overnight incubation at 25 °C.

For $^{15}$N-labeled proteins, the procedure was as described for the doubly labeled protein, but the M9 medium contained 0.4% (w/v) glucose instead of $^{13}$C glucose.

Purification of all isotope-labeled proteins was performed as described for their unlabeled counterparts.

**NMR analysis.** All NMR experiments were performed at $T = 298$ K on a Bruker Avance NEO 800 MHz spectrometer equipped with cryogenic triple resonance TCI or quadruple resonance QCI probes. Datasets were processed using NMRPipe (version 10.1) and analyzed by NMRViewJ (version 8.0a27)[58,59]. Measurements were performed on singly $^{15}$N-labeled versions of NEDD8, pNEDD8, pUb, and $^{nh}$pUb, if not explicitly stated otherwise. All samples were prepared in 20 mM HEPES pH 7.4 with 150 mM NaCl and 5% (v/v) D$_2$O using concentrations of 260 µM (NEDD8), 80 µM (pNEDD8), 25 µM (pUb) and 20 µM ($^{nh}$pUb). Chemical shifts deposited in the Biological Magnetic Resonance Bank (BMRB) were used for backbone resonance assignment of NEDD8 (entry 10062 [https:// bmrb.io/data_library/summary/index.php?bmrbId=10062] for a His-tagged version of NEDD8) and pUb (entries 36082 [https://bmrb.io/data_library/ summary/index.php?bmrbId=36082] for the relaxed state and 36081 [https:// bmrb.io/data_library/summary/index.php?bmrbId=36081] for the retracted state)[21,60]. The remaining ambiguities regarding the assignment of NEDD8 were cleared by three-dimensional $^{15}$N-edited NOESY-HSQC (120 ms mixing time) and TOCSY-HSQC spectroscopy (80 ms mixing time). De novo backbone assignment of pNEDD8 was accomplished on basis of a doubly $^{13}$C/$^{15}$N-labeled sample with a concentration of 320 µM in the buffer described above. For this purpose, the three-dimensional TROSY-based triple resonance experiments HNCA, HNCO, HN(CO)

CACB, and HNCACB were conducted. Due to moderate cross-peak shifts, most of the assignment of pUb could be transferred directly to $^{nh}$pUb by visual estimate without further experiments.

Weighted chemical shift perturbations were calculated according to the following equation[61]:

$$\triangle\omega = \sqrt{\frac{(\Delta^1 H)^2 + \frac{1}{25}(\Delta^{15}N)^2}{2}}, \quad (1)$$

where $\Delta^1 H$ is the change in proton and $\Delta^{15}N$ is the change in nitrogen dimension between corresponding peaks.

To quantify the position of the conformational equilibrium between the relaxed and retracted state of pNEDD8, pUb, and $^{nh}$pUb, the fraction of the retracted state population, $f_n$, was determined from relative signal heights in the $^1$H-$^{15}$N HSQC spectrum by using

$$f_n = \frac{I_n}{I_m + I_n}, \quad (2)$$

where $I_m$ and $I_n$ are the signal heights of associated peaks representing the relaxed and retracted state population, respectively[20]: For each construct, 17 sets of well-separated cross-peaks were selected to calculate the mean (pNEDD8: L2, V5, K6, I13, I15, E18, I26, V30, I44, Y45, K48, E53, T55, K60, L62, L67, H68; pUb: I3, V5, K6, T14, L15, E18, K27, D32, I44, F45, Q49, R54, T55, D58, Q62, T66, L69; $^{nh}$pUb: I3, V5, K6, T14, L15, E18, K27, D32, I44, F45, L50, R54, T55, D58, Q62, L69, L71).

ZZ exchange experiments with mixing times of $t_{mix} = 20, 40, 60,$ and 100 ms were performed on a doubly $^{13}$C/$^{15}$N-labeled sample of pNEDD8 to identify the time scale of conformational exchange between the relaxed and retracted state[62]. The interconversion rate, $k_{conf}$, was determined by fitting the signal heights of auto- and cross-peaks of six selected residues (L2, I13, Y45, T55, L67, H68) that were well resolved and did not show any overlapping to Eq. (3)[63]:

$$\frac{I_{AB}(t_{mix})I_{BA}(t_{mix})}{I_{AA}(t_{mix})I_{BB}(t_{mix}) - I_{AB}(t_{mix})I_{BA}(t_{mix})} = k_{conf}^2 t_{mix}^2, \quad (3)$$

where $A$ and $B$ stand for the two conformations and $I_{AA}$ and $I_{BB}$ are the corresponding auto-peak signal heights and $I_{AB}$ and $I_{BA}$ are the corresponding cross-peak signal heights.

Secondary structural elements in the relaxed and retracted state conformation of pNEDD8 as well as dihedral angles ($\psi$, $\varphi$) were determined by TALOS-N[29] and chemical shift indices were calculated by means of the CSI 3.0 web server[64] by including HN, N, CA, and CO chemical shifts.

**Cell culture and lysate preparation.** HEK293T (ATCC) and SHSY5Y (ATCC) cells were cultured in Dulbecco's Modified Eagle's Medium (DMEM, Thermo Fisher) with 10% fetal bovine serum (FBS, Sigma).

Cell pellets were resuspended in lysis buffer (1x PBS, 2 mM MgCl$_2$, 1 mM DTT, 1 mg/ml Aprotinin/Leupeptin, 1 mg/ml Pefabloc) and lysed by sonication. After the clearance of the lysate by centrifugation (21,000× g, 30 min, 4 °C), the total protein concentration was determined with the BCA Protein Assay Kit (Thermo Fisher).

**Affinity enrichment.** For affinity enrichment of interacting proteins, 25 µg of each Strep-Ub / NEDD8 variant was incubated with 2.5 mg cell lysate (HEK293T or SHSY5Y) for 10 min on ice. After equilibration of 50 µl Strep-Tactin Superflow (iba) bead-slurry with three times 200 µl lysis buffer (1x PBS, 2 mM MgCl$_2$, 1 mM DTT, 1 mg/ml Aprotinin/ Leupeptin, 1 mg/ml Pefabloc), the mixture was added to the beads. Incubation at 4 °C for 3 h in an overhead shaker was followed by 5 washing steps with 200 µl lysis buffer each. Bound proteins were eluted by incubating the beads for 5 min on ice with two times 150 µl and one time 100 µl elution buffer (100 mM Tris-HCl pH 8.0, 150 mM NaCl, 1 mM EDTA, 2.5 mM desthiobiotin). The eluates were combined and subjected to freeze-drying. Each experiment was performed in triplicates. Note that since cell lysis was performed in the absence of nonionic detergents followed by high-speed centrifugation, it is likely that mitochondria/mitochondria-associated proteins are present only in low abundance in the cell lysate and are thus not or only poorly detectable.

The freeze-dried eluate was resuspended in 100 µl 8 M urea, reduced with 5 mM TCEP for 30 min at 37 °C, and alkylated with 10 mM iodoacetamide for 30 min at room temperature. After dilution to 1 M urea with 50 mM NH$_4$HCO$_3$, 5 µg trypsin were added and proteins were digested for 20 h at 37 °C. The digested samples were freeze-dried and kept at −20 °C until they were measured. Prior to MS measurement, the tryptic peptides were dissolved in 50 µl 0.1% TFA in H$_2$O, acidified with 10% TFA, and desalted using ZipTip C18$_\mu$.

For Western blot analysis, affinity enrichment was performed as described above. The combined eluates were concentrated to 50 µl (SpeedVac, Thermo Fisher), 10% of which were applied to SDS-PAGE followed by Western blot analysis using antibodies specific for the respective protein (anti- HDAC6 (1:1,000; #7612 S, Cell Signaling Technologies), anti-USP16 (1:1,250; #HPA021140, Sigma), anti-UBA1 (1:1,000; #4891 S, Cell Signaling Technologies) anti-UBE2M (1:1,000; #4913 S, Cell Signaling Technologies), anti-HSPA8 (1:1,000; #PA5-27337, Thermo Fisher)) and as secondary antibody anti-rabbit HRP (1:15,000; #111-035-003, Jackson ImmunoResearch). "Input" represents 1% of the HEK293T cell lysate used for affinity enrichment.

**Ectopic expression of HA-His-NEDD8.** HEK293T cells were seeded in 6 cm dishes with Dulbecco's Modified Eagle's Medium (DMEM, Thermo Fisher) supplemented with 10% fetal bovine serum (FBS, Sigma) 24 h prior to transfection. 4 µg of pcDNA HA-His-NEDD8 or empty vector as control were transfected using Lipofectamine 2000 (Thermo Fisher) according to the manufacturer's instructions. 24 h after transfection, cells were treated with either 30 µM CCCP in DMSO for 6 h or 20 µM Etoposide (TCI) in DMSO for 16 h to induce mitochondrial stress and DNA double-strand breaks, respectively. Pure DMSO was used as a control.

**Affinity enrichment of HA-His-NEDD8.** Cells were harvested in 1.5 ml ice cold 1x PBS. Prior to centrifugation, 375 µl were removed and pelleted separately as input samples for Western blot analysis. The latter pellet was resuspended in 84 µl 2x loading buffer (5% (w/v) SDS, 25% (v/v) glycerol, 150 mM Tris-HCl pH 6.8, 0.01% (w/v) bromophenol blue), heated to 95 °C for 5 min and the cells lysed by sonication. The total protein concentration was determined with the BCA Protein Assay Kit (Thermo Fisher), and β-mercaptoethanol was added to a final concentration of 0.7 M. The main cell pellet was resuspended in 500 µl lysis buffer (6 M guanidinium hydrochloride, 100 mM Na$_2$HPO$_4$ pH 8.0, 10 mM imidazole, 10 mM β-mercaptoethanol, 1 mg/ml Aprotinin/Leupeptin, 5 mM β-glycerophosphate, 2 mM NaF), incubated on ice for 1 h and cells lysed by sonication. Cell lysate was precleared by incubation with 150 µl equilibrated Sepharose 4B (GE) beads for 1 h at 4 °C. After centrifugation (11,688× g, 5 min, RT), the supernatants were transferred to 50 µl equilibrated Nickel-NTA beads (Qiagen) and incubated at 4 °C overnight. After centrifugation (11,688× g, 5 min, RT), the supernatant was removed, and the beads were washed with lysis buffer followed by a 1:4 mixture of lysis and wash buffer and a final washing step with pure wash buffer (50 mM Tris-HCl pH 6.8, 20 mM imidazole). Bound HA-His-NEDD8 and neddylated proteins were eluted with 30 µl elution buffer (2x loading buffer, 200 mM imidazole). 20% of the elution were used for Western blot analysis together with the input samples (anti-HA (1:2,000; #901533, Biolegend), anti-Tubulin (1:5,000; #ab7291, abcam)) and as secondary antibody anti-mouse HRP (1:15,000; #115-035-062, Jackson ImmunoResearch). 80% of the elution were used for in-gel tryptic digest according to Shevchenko et al.[65]. Prior to MS measurement, the tryptic peptides were dissolved in 40 µl 0.1% TFA in H$_2$O, acidified with 10% TFA, and desalted using Pierce™ C18 Spin Tips (Thermo Fisher).

**Shotgun proteomics of the Strep-Ub/NEDD8 affinity enrichment.** The protein digests were analyzed on a Q-Exactive HF mass spectrometer (Thermo Fisher Scientific, Bremen, Germany) interfaced with an Easy-nLC 1200 nanoflow liquid chromatography system (Thermo Scientific, Odense, Denmark). The digests were reconstituted in 0.1% formic acid and loaded onto the analytical column (75 µm × 15 cm). Peptides were resolved at a flow rate of 300 nL/min using a linear gradient of 6−40% solvent B (0.1% formic acid in 80% acetonitrile) over 165 min. Data-dependent acquisition with full scans in 350−1500 m/z range was carried out using the Orbitrap mass analyser at a mass resolution of 60,000 at 200 m/z. The 10 most intense precursor ions were selected for further fragmentation. Only peptides with charge states 2−6 were used and dynamic exclusion was set to 30 s. Precursor ions were fragmented using higher-energy collision dissociation (HCD) with a normalised collision energy (NCE) set to 28%. Fragment ion spectra were recorded at a resolution of 15,000. Each of the three biological replicates was measured in technical duplicates.

**Targeted proteomics of the HA-His-NEDD8 affinity enrichment.** The protein digests were analyzed with a Q-Exactive HF mass spectrometer (Thermo Fisher) interfaced with an Easy-nLC 1200 nanoflow liquid chromatography system (Thermo Fisher). The digests were reconstituted in 0.1% formic acid and loaded onto the analytical column (75 µm × 15 cm). Peptides were resolved at a flow rate of 300 nL/min using a linear gradient of 6−40% solvent B (0.1% formic acid in 80% acetonitrile) over 45 min. Selected peptide ion monitoring with full scans in the 350−1500 m/z range was carried out using the Orbitrap mass analyser at a mass resolution of 120,000 at 200 m/z. The target precursor ions were selected for further fragmentation. Only peptides with charge states 2−6 were used and dynamic exclusion was set to 30 s. Precursor ions were fragmented using higher-energy collision dissociation (HCD) with a normalised collision energy (NCE) set to 28%. Fragment ion spectra were recorded at a resolution of 30,000. Each of the biological replicates was measured in technical duplicates or triplicates. See Source Data for the peptide target list.

**Data analysis of Strep-Ub/NEDD8 affinity enrichment.** For label-free quantification, the raw files from LC-MS/MS measurements were analyzed using Max-Quant (version 1.6.8) with default settings and match between runs and label-free quantification (LFQ) (minimum ratio count two) enabled[66,67]. Oxidation (M) and acetylation (N-terminus) were included as variable modifications and Carbamidomethyl (C) as a fixed modifications with a maximum of five modifications per peptide (default settings). The digestion mode was set to Trypsin/P, which cleaves after lysine and arginine also if a proline follows, and a maximum of two missed cleavages was allowed (default settings). The minimal peptide length was set to seven with a mass tolerance of 4.5 ppm and 20 ppm for parent ions and fragment ions, respectively (default settings). The protein- and peptide-level FDR was set to

0.01 (default settings). For protein identification, the human reference proteome downloaded from the UniProt database (download date: 2018-02-22) and the integrated database of common contaminants were used. For further data processing, Perseus software (version 1.6.10.50) was used[68]. Identified proteins were filtered for reverse hits and common contaminants. LFQ intensities were log2 transformed and the proteins filtered to be detected in at least 4 replicates among 6 replicate experiments. Missing values were imputed from a normal distribution (width = 0.3 and shift = 1.2) based on the assumption that these proteins were just below the detection limit. Significantly enriched proteins were identified by an ANOVA test (FDR = 0.02, S0 = 1), and the rows Z-score normalized. After averaging all replicates of the respective groups, the proteins with values higher in the bead control sample than in any of the other samples were filtered out. Enriched proteins were clustered on correlation and plotted as heatmaps. The addition of annotations (GO annotations, KEGG, Pfam) and identification of enriched terms were done in Perseus. DUBs, E2s, E3s, and proteasome subunits were identified by comparison with the UUCD database[69], DUB inventory[70], human E3 database[71], and the KEGG proteasome (Pathway hsa03050).

**Data analysis of HA-His-NEDD8 affinity enrichment**. MS spectra were analyzed using Proteome Discoverer software version 1.4 (Thermo Fisher). The fragmentation spectra were searched against the human reference proteome from the Swissprot database (download date: 2020-05-26) with the MASCOT search engine (with the following settings applied: protease cleavage, trypsin with maximal one missed cleavage site; fixed modification, Carbamidomethyl (C); variable modifications, oxidation (M) and phosphorylation (S); precursor mass tolerance, 5 ppm; fragment mass tolerance, 10 mmu; precursor mass, between 350 Da (min.) and 5000 Da (max.); peptide-level FDR, 0.01); peak area calculation was performed using the Event and Precursor Ions Area Detector nodes (mass precision 2 ppm). The calculated peak area data were further processed using Graph Pad Prism 6. The mean of NEDD8 peptides that do not comprise S65 were used for baseline correction (division by baseline). After normalization by setting the sum of S65 and pS65 peptide area to 100%, the relative intensities of the pS65-containing NEDD8 peptide present under the different conditions were plotted.

**HSP70 ATPase activity measurement**. 1 μM N-terminally His$_6$-tagged HSP70 ATPase domain (residues 1–402, including the ATPase activity stimulating linker domain) was incubated with 10 μM Ub, pUb, $^{nh}$pUb, NEDD8, pNEDD8, $^{nh}$pNEDD8, NEDD8 LIA or pNEDD8 LIA in assay buffer (100 mM Tris-HCl pH 7.5, 20 mM KCl, 6 mM MgCl$_2$) for 1 h on ice. The reaction was started by the addition of 0.5 μl of 11 mM ATP to reach a final reaction volume of 30 μl. After incubation at 37 °C for the times indicated, 25 μl of the reaction mixture was added to 75 μl of green malachite reagent (ATPase/GTPase activity assay kit, SIGMA) in a 96-well plate and incubated at room temperature for 20 min. The absorbance was measured at 650 nm with a TECAN infinite F500 plate reader. Samples lacking HSP70 but otherwise treated identically were used to correct for non-enzymatic ATP hydrolysis. For the sample without ATP, ultrapure water was added instead of ATP. A phosphate standard was used to calculate absolute values.

**Reporting summary**. Further information on research design is available in the Nature Research Reporting Summary linked to this article.

## Data availability

The data that support this study are available from the corresponding authors upon reasonable request. NMR resonance assignments have been deposited in the Biological Magnetic Resonance Data Bank with the following accession numbers: 50466 (NEDD8), 50467 (pNEDD8 relaxed state), and 50468 (pNEDD8 retracted state). Preexisting NMR resonance assignments used in this study are available under the accession codes 10062 (His-tagged NEDD8), 36082 (pUb relaxed state), and 36081 (pUb retracted state). The NEDD8 crystal structure data, the NMR solution structure of NEDD8, and the NMR solution structure of ubiquitin used in this study are available in the Protein Data Bank under the accession codes 1NDD, 2KO3, and 1D3Z, respectively. The mass spectrometry proteomics data have been deposited to the ProteomeXchange Consortium (http://proteomecentral.proteomexchange.org) via the PRIDE partner repository[72] with the dataset identifier PXD021143 for the affinity enrichment data and the dataset identifier PXD027477 for the pNEDD8 analysis data. Source data are provided with this paper.

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

## Acknowledgements

We thank Andreas Marquardt of the Proteomics Center at the University of Konstanz for assistance with the mass spectrometric experiments, Silke Büstorf for excellent technical assistance, Eva Höllmüller for valuable input on AE-MS experiments, and Simon Kienle for discussion and critical feedback on the project. This work was supported by the Konstanz Research School Chemical Biology (KoRS-CB) and by the DFG (SFB969, Projects B3, and B9).

## Author contributions

K.S., M.K., A.M., and M.S. conceived and supervised the study. K.S. designed, performed, and analyzed experiments and purified proteins. K.S. and J.W. carried out ubiquitylation assays. T.S. carried out NMR experiments and analyzed them with M.K. K.S., T.S., and M.S. wrote the paper. All authors provided critical feedback on the paper.

## Funding

## Competing interests

The authors declare no competing interests.

## Additional information

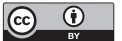

