## [Peer Review File · Nature Communications]

REVIEWER COMMENTS

Reviewer #1 (Remarks to the Author):

The manuscript by Stuber et al. describes the structural and functional characterization of phosphorylation for the ubiquitin-like molecule NEDD8. Based on proteomics studies that identified phosphorylation sites on NEDD8 in cells, the authors investigated NEDD8 phosphorylation on S65. This is a conserved residue between NEDD8 and Ubiquitin and critically S65-phospho-Ubiquitin has an established role for the mitochondrial activation of Parkin E3-ligase. The authors performed biochemical analysis showing that S65-phospho-NEDD8 can also activate Parkin in vitro and detailed NMR analysis characterizing distinct conformations of the unmodified and phospho-state of NEDD8. This part of the manuscript shows the effect of NEDD8 phosphorylation at structural and biochemical level. The authors then applied a chemical-like biology approach to identify by mass-spectrometry based proteomics, binders for the phospho-state of NEDD8. The data indicate that the phospho-NEDD8 has distinct proteome compared to its closest homologue phospho-Ubiquitin and critically from the unmodified NEDD8. Based on these data the authors show that phospho-NEDD8 is a better activator of the HSP70 ATPase activity compared to unmodified NEDD8.

In general, this is a very well performed study, which provides interesting new insights and concepts on how post-translational modifications of the Ubl NEDD8 affect NEDD8 function.

I think the proteomics analysis is great and even if it appears that it has not gone too deep (based on known Ub binders) it clearly provides significant new information on NEDD8 interactions and the potential role of phosphorylation as modulation signal.

My major concern is on the data presented for parkin activation. The biochemical analysis could be improved:

- Use of tagged unmodified Ub to discriminate from the pUb. Currently it is not possible to compare the effect of pUb to the effect of pNEDD8 on parkin activation as from fig. 1d it is concluded that the used anti-Ub antibody recognizes both forms of Ub.
- Use an anti-parkin western in these experiments.
- Use of the Ubl Δ -Parkin mutant which is a better "reader" of the effect of pUb.
- Use of S65A mutants for pUb/pNEDD8. In Fig. 1c any explanation for the difference in size for pUb and pNEDD8?

I feel that for this particular finding to have a significant impact in the field some sort of biological insights are required. For example, show that NEDD8 is phosphorylated at S65 upon mitochondrial stress? (This maybe challenging as it requires phospho-specific antibodies-maybe purification of NEDD8DGG under denaturing conditions and use of phos-tag?). Effect of NEDD8 knockdown on parkin activation, with potentially co-expression of NEDD8 siRNA resistant phospho-mutants? Effect of NEDD8 siRNA on parkin activation has been tested in the study by Um et al., 2012, J. Neurosci Res. that should be discussed.

NEDD8 phosphorylation as potential mechanism for control of NEDD8 function may prove critical. The authors could provide a broader discussion of the reported functions of NEDD8 particularly upon stress conditions and the implications of their findings.

The role of NUB1/NUB1L in targeting NEDD8 proteins for degradation is not clear. Has been challenged by more recent studies of the Huabo Su laboratory and Maghames et al. 2018-NatCom. The presented proteomics studies do not identify proteasomal subunits as NEDD8 interacting proteins which was expected based on the initial proposed role of NUB1 in proteasomal degradation of NEDD8 conjugates.

Fig. 1C. The presence of CIP should be indicated in the western blot

Sup. Fig. S5. The effect of nhpNEDD8 on parkin activation is not shown.

Reviewer #2 (Remarks to the Author):

Stuber and colleagues report that NEDD8 can be phosphorylated on serine 65 by the ubiquitin-kinase PINK1 and act as an allosteric activator of the ubiquitin ligase Parkin. The paper is roughly divided into three sections: 1) characterization of NEDD8 phosphorylation and its activation of Parkin, 2) an NMR study of phosphorylated NEDD8 (pNEDD8) showing two conformations as observed with phosphorylated ubiquitin (pUb), and 3) an MS interactome study that found pNEDD8 binds and activates HSP70.

While potentially interesting, the study is overly ambitious and premature. The experiments to show the physiological relevance of NEDD8 phosphorylation are not convincing. A second issue is that the three sections are only loosely connected. It is unclear how the alternative pNEDD8 conformations impact Parkin activation or HSP70 binding. The MS interactome analysis doesn't provide insight into NEDD8's supposed involvement with PINK1 and Parkin.

Major points

Previous work from Miratul Muqit has shown that PINK1 can phosphorylate NEDD8 (Kazlauskaitė et al, *Biochem J*, 2015). The amount phosphorylated was small but given the close sequence similarity between NEDD8 and ubiquitin, it isn't a surprising result. Here, Stuber and colleagues observe somewhat more phosphorylation of NEDD8 but it is still less than with ubiquitin as a substrate. The claim that it is with an "efficiency similar to that of Ub phosphorylation" is incorrect. At 1.5 h, ubiquitin is 50% phosphorylated while NEDD8 never gets close to 50%. The claim that NEDD8 is a "bona fide" substrate is premature. The authors don't measure NEDD8 phosphorylation in vivo and don't show an increase by stimuli that increase PINK1 levels.

There are a few proteomic papers that report phosphoS65 NEDD8 but there is no evidence that it is physiological. The phosphorylation reported of NEDD8 by DCNL5 was on serine 41. More generally, NEDD8 is largely present in the nucleus rather than in cytoplasm so it is unclear how it can be phosphorylated by the mitochondrial kinase PINK1. Reference 37 in the manuscript (Balasubramaniam, M. et al., *J. Neuroinflammation*, 2019) claims that NEDD8 is translocated from the nucleus to bind Parkin in IL-1 β -treated neurons but they didn't report NEDD8 phosphorylation.

The key result in the paper is the activation of parkin by pNEDD8. The authors use a GST-Parkin autoubiquitylation assay analyzed by Coomassie staining to observe ubiquitin-protein conjugates and western blotting for ubiquitin. The conditions of the assay are a bit unusual with a relatively small amount of GST-parkin and significant basal activity but the main issue is that the Coomassie gels show no evidence of activation. In Fig 1d, the intensity of unmodified Parkin doesn't change in the pNEDD8-treated samples. In contrast, pUb addition leads to a decrease of unmodified Parkin at ~80 kD, and a smear of polyubiquitylated protein at the top of the gel. Parkin activity can also be seen in the formation of an Ub-E2 conjugate at ~ 30kD. The results with Coomassie staining in Fig 1e are similar – activity with pUb, little or none with pNEDD8.

As an alternative, the authors use western blots with anti-ubiquitin but there are a couple of issues that cast doubt on their reliability. Firstly, the westerns show significant ligase activity in the absence of either pUb or pNEDD8. Secondly, it is very odd that the authors load twice as much material on the gel for the western blots as for Coomassie staining. My interpretation is that a small amount of Parkin in their samples is partially denatured and responsible for the activity without pUb. Parkin activity can be partially activated simply by mild heating (<https://pubmed.ncbi.nlm.nih.gov/23861750/>). The absence of changes in the Coomassie gels suggests that only a small fraction of the Parkin molecules are activated in the presence of pNEDD8. The bottom line is that the evidence for Parkin activation by

pNEDD8 is not compelling.

The phosphoserine binding sites on Parkin are well-known and mutations that block activation by pUb and pUbl (H302A or K211N) are well characterized. To validate their Parkin autoubiquitination assays, the authors could test activation with these Parkin mutants. The results would prove that pNEDD8 is working through direct binding of Parkin and have the additional value of potentially identifying the binding site.

The NMR evidence for two conformations of phosphorylated Ub and NEDD8 was generally satisfactory but the depth of the structural analysis was limited. I found it surprising that although the authors claim NEDD8 is a good substrate of PINK1, they needed to use genetic code-expansion to directly incorporate phosphoserine and obtain sufficient, homogeneously phosphorylated protein. However, in spite of this, the pNEDD8 NMR samples still contained significant amounts of unphosphorylated protein which limited the analysis to the interpretation of chemical shifts.

The last section of the manuscript was the interactome analysis. This naturally generated a lot of data but the robustness and completeness of the results were undercut by the absence of Parkin among the hits for pUb binding. Parkin binds pUb with nanomolar affinity. The authors need to explain why it was absent.

The MS analysis identified numerous interactions of pNEDD8 with protein chaperones such as HSP70. The physiological relevance is again unclear but could be the result of partial unfolding or misfolding of pNEDD8. In agreement with chaperone binding, the authors observed stimulation of HSP70 ATP-activity by pNEDD8. But, it was hard to see how the interactome results were related to the PINK1-Parkin pathway or how they were evidence of biological relevance of NEDD8 Ser65 phosphorylation.

Minor points:

The use of the term ubiquitin-like modifier to refer to NEDD8 would be less confusing than "Ubl" which is also used for the Ubl domain of Parkin.

The volume of the autoubiquitination assay should be given so that the pUb and pNEDD8 concentrations can be calculated. The amounts are given as weights compared to molar concentrations for the other components.

In Fig 4f, the HEK293T clusters 5 and 6 seem to be missing.

Reviewer #3 (Remarks to the Author):

In their study entitled "Structural and functional consequences of NEDD8 phosphorylation", the authors report and dissect the structural consequences of the phosphorylation of NEDD8, a Ub-like protein, in Ser65. Phosphorylated NEDD8 activates the Ub ligase Parkin. In addition, the authors study the interactomes of Ub, pUb, NEDD8, pNEDD8, and also of non-hydrolyzable pUb and pNEDD8. Analysis of these interactomes unveils novel cellular functions for pUb and pNEDD8 beyond Parkin activation and mitophagy. In particular, the authors found that pNEDD8 enhances the activity of HSP70 in vitro.

The experiments are well designed, the overall quality of the data is high, the information is clear, and the manuscript is well written.

Although describing very interesting findings as the phosphorylation of NEDD8, its role in Parkin activation, its structural consequences, and the differential interactome of the different forms of Ub and NEDD8, there are some aspects that, in my opinion, would require further investigation to be

published in Nature Communications.

Major comments:

1. Lack of biology. You would expect some assays showing the biological consequences of NEDD8 phosphorylation in addition to the reported in vitro assays.
 - 1.1. Ubiquitin is phosphorylated on Ser65 upon mitochondrial depolarisation leading to Parkin activation and mitophagy. I would suggest to the authors to induce mitochondrial depolarisation (i.e. with carbonyl cyanide 3-chlorophenylhydrazone (CCCP)) and look at presence of phospho-NEDD8 using the Phos-tag gels followed by Immunoblotting against NEDD8 (I do not think there is an antibody against pNEDD8, but Immunoblotting against it would definitely be a better option).
 - 1.2. The authors suggest that DNA damage induces NEDD8 phosphorylation - they could do the same as in point 1.1. using DNA damage agents to prove this claim.
 - 1.2. What are the consequences in Mitophagy after introducing a phosphodeficient mutant of NEDD8 (S65A) in NEDD8 KO cells? This model could be used to study the consequences of not having phosphorylated NEDD8 in the additional signaling pathways identified in the interactome analysis such as autophagy, DNA damage, apoptosome formation, etc.
2. Validation of the interactome data using an orthogonal method such as Immunoblotting after enrichment with the described baits or, alternatively, performing Co-Immunoprecipitations using existing Ub, phospho-Ub, and NEDD8 antibodies.

Minor points:

1. Can you prove that the NEDD8-His variant does not conjugate to proteins using an in vitro NEDDylation assay?
2. To preserve mitochondrial proteins the centrifugation speed during cell lysis should be lower than the 21,000 g the authors used here. This might be the reason the usual suspects were not identified in this interactome study (Parkin, USP30, and other mitochondrial proteins such as VDACS).
3. A more elaborate description of the stress pathways identified would be beneficial for the quality of the manuscript.

Point-by-point response to reviewers' comments on "Structural and functional consequences of NEDD8 phosphorylation" (NCOMMS-20-39525-T)

Reviewer #1

The manuscript by Stuber et al. describes the structural and functional characterization of phosphorylation for the ubiquitin-like molecule NEDD8. Based on proteomics studies that identified phosphorylation sites on NEDD8 in cells, the authors investigated NEDD8 phosphorylation on S65. This is a conserved residue between NEDD8 and Ubiquitin and critically S65-phospho-Ubiquitin has an established role for the mitochondrial activation of Parkin E3-ligase. The authors performed biochemical analysis showing that S65-phospho-NEDD8 can also activate Parkin in vitro and detailed NMR analysis characterizing distinct conformations of the unmodified and phospho-state of NEDD8. This part of the manuscript shows the effect of NEDD8 phosphorylation at structural and biochemical level. The authors then applied a chemical-like biology approach to identify by mass-spectrometry based proteomics, binders for the phospho-state of NEDD8. The data indicate that the phospho-NEDD8 has distinct proteome compared to its closest homologue phospho-Ubiquitin and critically from the unmodified NEDD8. Based on these data the authors show that phospho-NEDD8 is a better activator of the HSP70 ATPase activity compared to unmodified NEDD8.

In general, this is a very well performed study, which provides interesting new insights and concepts on how post-translational modifications of the Ubl NEDD8 affect NEDD8 function.

I think the proteomics analysis is great and even if it appears that it has not gone too deep (based on known Ub binders) it clearly provides significant new information on NEDD8 interactions and the potential role of phosphorylation as modulation signal.

My major concern is on the data presented for parkin activation. The biochemical analysis could be improved:

- Use of tagged unmodified Ub to discriminate from the pUb. Currently it is not possible to compare the effect of pUb to the effect of pNEDD8 on parkin activation as from fig. 1d it is concluded that the used anti-Ub antibody recognizes both forms of Ub.

We agree with the reviewer that the experiment shown in Fig. 1d does not allow to compare the effects of pUb and pNEDD8 on Parkin activation. This was one of the reasons for performing experiments with C-terminally His-tagged versions of pUb and pNEDD8, since these cannot be activated and, thus, cannot be attached to Parkin or form Ub chains. Thus, in the experiment to Fig. 1e (as well as in respective lanes of the new Fig. 1d), only Parkin covalently attached to unmodified Ub is decorated in the Western blot analysis by the anti-Ub antibody. We hope that the reviewer agrees that this allows to compare the effects of pUb and pNEDD8. A respective sentence has been included in the revised manuscript (page 5, 1st paragraph).

- Use an anti-parkin western in these experiments.

In response to the comments of reviewer 2, we performed additional experiments with the Parkin mutant H302A (new Fig. 1d). The respective experiment was not only analyzed by Coomassie staining and anti-Ub Western blot but also by anti-Parkin Western blot. However, as can be seen below (Fig. 1 for reviewer), the high molecular mass forms of Parkin, representing Parkin modified by ubiquitin chains and/or multiple mono-ubiquitylated Parkin, are poorly detected by the anti-Parkin antibody. This is explained by the fact that such Parkin forms display exactly one epitope that can be bound by only one anti-Parkin antibody, while they display multiple epitopes for the anti-Ub antibodies (and thus are decorated by multiple anti-Ub antibodies). Since the anti-Parkin analysis does not provide any additional information to the Coomassie staining (see also our response to reviewer 2), we decided not to include these data in the revised manuscript. We would also like to point out that anti-Parkin Western blots are not provided in other reports on pUb-mediated Parkin activation as well (e.g. Wauer et al., EMBO J 34, pp. 307; Kumar et al., EMBO J 34, pp. 2506; Condos et al., EMBO J 37, e100014).

Fig. 1 for reviewer. Autoubiquitylation was performed in the presence or absence of the Ub/NEDD8 variants indicated. The ratio of Ub to the different Ub/NEDD8 variants was 9:1. Reactions were stopped after 60 min and analyzed by SDS-PAGE followed by Western blot analysis with an anti-Ub antibody (upper panel) or with an anti-Parkin antibody (lower panel).

- Use of the Ub Δ Parkin mutant which is a better “reader” of the effect of pUb.

As suggested by the reviewer, we performed experiments with a Parkin mutant devoid of the Ub-like domain (Fig. 2 for reviewer). However, since the results obtained do not significantly differ from those obtained with full-length Parkin (Fig. 1d), we decided not to include this experiment in the revised manuscript.

Fig. 2 for reviewer. Parkin autoubiquitylation was performed in the presence or absence of the Ub/NEDD8 variants indicated. The ratio of Ub to the different Ub/NEDD8 variants was 9:1. Reactions were stopped after 60 min and analyzed by SDS-PAGE followed by Western blot analysis with an anti-Ub antibody (upper panels) or by Coomassie blue staining (lower panel).

- Use of S65A mutants for pUb/pNEDD8. In Fig. 1c any explanation for the difference in size for pUb and pNEDD8?

As all the proteins used in the ubiquitylation experiments were expressed in bacteria and thus Ub and NEDD8 are not modified at S65 (this was also proven by mass spec analysis, Fig. S1b), we decided not to perform experiments with S65A mutants, as in our opinion such experiments would not provide any additional information (unless we did not understand the reviewer's intention).

Ub and NEDD8 are similar but not identical in their amino acid sequence and, thus, it is not surprising that phosphorylated forms migrate differently in a Phos-tag gel. pUb and pNEDD8 were analyzed on the same gel (Fig. 1c) as shown below (Fig. 3 for reviewer). For reasons of space, we decided not to include the entire gel in the manuscript. We would also like to point out that the mass spec analysis of pNEDD8 demonstrated that phosphoserine was exclusively incorporated at position 65 (Figs. S1a, S1b).

Fig. 3 for reviewer. pUb and pNEDD8 were incubated in the absence (-) or presence (+) of alkaline phosphatase (CIP). Upon treatment, reaction products were analyzed by Phos-tag SDS-PAGE followed by Coomassie blue staining.

I feel that for this particular finding to have a significant impact in the field some sort of biological insights are required. For example, show that NEDD8 is phosphorylated at S65 upon mitochondrial stress? (This maybe challenging as it requires phospho-specific antibodies-maybe purification of NEDD8DGG under denaturing conditions and use of phos-tag?). Effect of NEDD8 knockdown on parkin activation, with potentially co-expression of NEDD8 siRNA resistant phospho-mutants? Effect of NEDD8 siRNA on parkin activation has been tested in the study by Um et al., 2012, J. Neurosci Res. that should be discussed.

Thank you for these helpful suggestions. Accordingly, we ectopically expressed an N-terminally His-tagged version of NEDD8 in cells, treated these with CCCP to induce mitochondrial stress, and purified NEDD8 under denaturing conditions. While this did not allow to detect pNEDD8 on a Phos tag gel (possibly because pNEDD8 levels were too low), we were able to show by mass spec analysis that CCCP stimulates phosphorylation of NEDD8 by ~1.7-fold (which appears to be similar to published data on Ub phosphorylation. In addition, total levels of pNEDD8 appear to be similar to pUb; ref. 11 of the revised manuscript). Moreover, the mass spec analysis allowed us to analyze the phosphorylation status of NEDD8 specifically at S65. These data are included in the revised manuscript as Fig. 1h and in Figs. S1d-e.

We feel that results obtained in NEDD8 knockdown experiments would be not informative or rather difficult to interpret, since pUb would still be present. However, it is not possible to simultaneously knockdown Ub expression and NEDD8 expression (as in contrast to NEDD8, Ub is not only involved in Parkin activation but is also required for ubiquitylation). Although we cannot draw conclusions on how much NEDD8 phosphorylation contributes to Parkin activation, we hope that the reviewer agrees that the observation that NEDD8 phosphorylation is stimulated by mitochondrial stress at least suggests that it is involved. In addition, we would like to emphasize that the main message of our manuscript is not that pNEDD8 plays a *critical* role in Parkin

activation (though it can contribute to it) but that pNEDD8 and pUb have clearly distinct structural and functional properties in comparison to unmodified NEDD8 and Ub.

Discussion of the data by Um et al. is now included (page 15, 1st paragraph).

NEDD8 phosphorylation as potential mechanism for control of NEDD8 function may prove critical. The authors could provide a broader discussion of the reported functions of NEDD8 particularly upon stress conditions and the implications of their findings.

We have added additional information on the reported functions of NEDD8 (page 15, 1st paragraph; page 17, 1st and 2nd paragraph) but did not go into much detail, due to space constraints. In addition, since pNEDD8 stimulates Hsp70 activity more prominently than unmodified NEDD8 and as suggested by reviewer 3, we determined the phosphorylation status of NEDD8 upon treatment of cells with the DNA damaging agent etoposide. Intriguingly, in contrast to mitochondrial stress this resulted in a decrease of NEDD8 phosphorylation (Fig. S6e-g). We discuss this new data in the manuscript in the context of the effect of pNEDD8 on Hsp70 activity (page 17, 1st paragraph). Although we can only speculate on the functional consequences of this differential phosphorylation, we are convinced that the notion that different stress stimuli have different effects on NEDD8 phosphorylation are of significant interest to the community.

The role of NUB1/NUB1L in targeting NEDD8 proteins for degradation is not clear. Has been challenged by more recent studies of the Huabo Su laboratory and Maghames et al. 2018-NatCom. The presented proteomics studies do not identify proteasomal subunits as NEDD8 interacting proteins which was expected based on the initial proposed role of NUB1 in proteasomal degradation of NEDD8 conjugates.

We thank the reviewer for pointing this out. A respective discussion of the data is now provided (page 14, 2nd paragraph).

Fig. 1C. The presence of CIP should be indicated in the western blot

We apologize for not having done this in the first place. The migration position of CIP on this Coomassie-stained gel is now indicated.

Sup. Fig. S5. The effect of nhpNEDD8 on parkin activation is not shown.

The effect of nhpNEDD8, which is similar to pNEDD8, is now provided in the new Fig. S5.

Reviewer #2

Stuber and colleagues report that NEDD8 can be phosphorylated on serine 65 by the ubiquitin-kinase PINK1 and act as an allosteric activator of the ubiquitin ligase Parkin. The paper is roughly divided into three sections: 1) characterization of NEDD8 phosphorylation and its activation of Parkin, 2) an NMR study of phosphorylated NEDD8 (pNEDD8) showing two conformations as observed with phosphorylated ubiquitin (pUb), and 3) an MS interactome study that found pNEDD8 binds and activates HSP70.

While potentially interesting, the study is overly ambitious and premature. The experiments to show the physiological relevance of NEDD8 phosphorylation are not convincing. A second issue is that the three sections are only loosely connected. It is unclear how the alternative pNEDD8 conformations impact Parkin activation or HSP70 binding. The MS interactome analysis doesn't provide insight into NEDD8's supposed involvement with PINK1 and Parkin.

We apologize that we have obviously failed to make the connection between the three sections sufficiently clear and that we apparently evoked the impression that the main message of our study is that pNEDD8 plays a *critical* role in Parkin activation. To address these issues, we have reorganized the abstract and provided additional information in-between the different Results sections to make their connection more obvious. In addition, the purpose of the interactome analysis was not to obtain more insight into the NEDD8-Parkin connection but to provide evidence that the role of pUb/pNEDD8 is not limited to Parkin activation but that these are involved in other processes as well. We hope that the reviewer agrees that our data are in strong support of the latter hypothesis.

Major points

Previous work from Miratul Muqit has shown that PINK1 can phosphorylate NEDD8 (Kazlauskaitė et al, Biochem J, 2015). The amount phosphorylated was small but given the close sequence similarity between NEDD8 and ubiquitin, it isn't a surprising result. Here, Stuber and colleagues observe somewhat more phosphorylation of NEDD8 but it is still less than with ubiquitin as a substrate. The claim that it is with an "efficiency similar to that of Ub phosphorylation" is incorrect. At 1.5 h, ubiquitin is 50% phosphorylated while NEDD8 never gets close to 50%. The claim that NEDD8 is a "bona fide" substrate is premature. The authors don't measure NEDD8 phosphorylation in vivo and don't show an increase by stimuli that increase PINK1 levels.

First, we would like to apologize that we failed to sufficiently recognize the data by Kazlauskaitė et al. in the original version of the manuscript (ref. 13 of the revised manuscript); this has been corrected in the revised version. Kazlauskaitė et al. observed that PINK1 can phosphorylate NEDD8 with very low efficiency, while quantification of our data indicates that PINK1 phosphorylates NEDD8 with an efficiency of approximately 50 percent of that of Ub phosphorylation (see below, Table 1 for reviewer). This is now indicated in the text of the revised manuscript (page 4, 2nd paragraph). We hope that the reviewer agrees that NEDD8 is a reasonably good substrate for PINK1 in vitro.

	t (h)	Band intensity		% phosphorylated
		unphosphorylated	phosphorylated	
Ub	0	6.77	0	0
	0.5	5.96	2.11	26
	1	4.99	3.4	41
	1.5	4.29	4.25	50
	2	3.93	4.32	52
NEDD8	0	7.9	0	0
	0.5	7.49	0.87	10
	1	6.98	1.5	18
	1.5	6.32	2.1	25
	2	5.82	2.18	27

Table 1 for reviewer. Quantification of the results shown in Fig. 1b.

To prove that NEDD8 is phosphorylated at S65 within cells, we ectopically expressed an N-terminally His-tagged version of NEDD8 in cells, treated these with CCCP to induce mitochondrial stress, and purified NEDD8 under denaturing conditions. By this, we were able to show by mass spec analysis that CCCP stimulates phosphorylation of S65 of NEDD8 by ~1.7-fold (which appears to be similar to published data on Ub phosphorylation. In addition total levels of pNEDD8 appear to be similar to pUb; ref. 11 of the revised manuscript). These data are included in the revised manuscript as Fig. 1h and in Figs. S1d-e. Although we cannot draw conclusions on how much NEDD8 phosphorylation contributes to Parkin activation, we hope that the reviewer agrees that the observation that NEDD8 phosphorylation is stimulated by mitochondrial stress at least suggests that it is involved.

There are a few proteomic papers that report phosphoS65 NEDD8 but there is no evidence that it is physiological. The phosphorylation reported of NEDD8 by DCNL5 was on serine 41. More generally, NEDD8 is largely present in the nucleus rather than in cytoplasm so it is unclear how it can be phosphorylated by the mitochondrial kinase PINK1. Reference 37 in the manuscript (Balasubramaniam, M. et al., J. Neuroinflammation, 2019) claims that NEDD8 is translocated from the nucleus to bind Parkin in IL-1 β -treated neurons but they didn't report NEDD8 phosphorylation.

Besides Balasubramaniam et al. (ref. 30 of the revised manuscript), Bailly et al. (ref. 31 of the revised manuscript) and Um et al. (ref. 17) showed that in addition to its role in the nucleus, NEDD8 has specific functions in the cytoplasm. Together with our mass spec data, this clearly indicates that NEDD8 can be phosphorylated in the cytoplasm. Furthermore, Balasubramaniam et al. presumably did not look for NEDD8 phosphorylation, so they did not report on it.

The key result in the paper is the activation of parkin by pNEDD8. The authors use a GST-Parkin autoubiquitylation assay analyzed by Coomassie staining to observe ubiquitin-protein conjugates and western blotting for ubiquitin. The conditions of the assay are a bit unusual with a relatively small amount of GST-parkin and significant basal activity but the main issue is that the Coomassie gels show no evidence of activation. In Fig 1d, the intensity of unmodified Parkin doesn't change in the pNEDD8-treated samples. In contrast, pUb addition leads to a decrease of unmodified

Parkin at ~80 kD, and a smear of polyubiquitylated protein at the top of the gel. Parkin activity can also be seen in the formation of an Ub-E2 conjugate at ~ 30kD. The results with Coomassie staining in Fig 1e are similar – activity with pUb, little or none with pNEDD8.

We respectfully disagree with the statement "the key result in the paper is the activation of parkin by pNEDD8". While this is presumably true for people working on Parkin, we would assume that the finding that pNEDD8/pUb have different interactomes and that these differ to their non-modified forms is more interesting to people working on NEDD8/Ub in general.

Chaugule et al. (EMBO J. 30, pp. 2853) reported that fusion of an N-terminal tag results in increased basal activity of Parkin. We did not comment on this issue in the manuscript, since we feel that this information does not add to the main message conveyed by the experiments shown. However, with due respect we do not agree with the notion that in the Coomassie-stained gels, the intensity of unmodified Parkin does not change in the presence of pNEDD8 and that a smear representing polyubiquitylated forms of Parkin cannot be observed. The decrease/increase in the intensity of unmodified Parkin/polyubiquitylated forms is admittedly less prominent than with pUb but visible (e.g. unmodified Parkin in Fig. 1d, and polyubiquitylated forms in Fig. 1e). Since this may be difficult to judge by eye-balling, we quantified the intensity of the band representing unmodified Parkin (Fig. 1d) and the intensity of the polyubiquitylated forms (Fig. 1d, Western blot analysis) (Table 2 for reviewer). This indicates that pNEDD8 is approximately 2-3fold less efficient in Parkin activation than pUb. This is now indicated in the text (page 5, 1st paragraph).

The ~30 kDa band is also visible in the pNEDD8 lane, though it is rather faint. We believe that this band represents a Ub oligomer rather than a UbCH7-Ub conjugate but in our opinion, this is not important for the conclusions of the manuscript.

		Coomassie Band of unmodified Parkin Band intensity normalized to Ub	anti-Ub Western Blot, Autoubiquitylated Parkin Band intensity normalized to Ub
Parkin	Ub	1	1
	10% pUb	0.68	7.37
	10% NEDD8	1.07	1.80
	10 % pNEDD8	0.92	3.91
H302A	Ub	1.00	1.00
	10% pUb	0.97	1.63
	10% NEDD8	0.96	0.96
	10 % pNEDD8	1.05	1.11
Parkin	10% Ub-His	1.00	1.00
	10% pUb-His	0.63	3.15
	10% NEDD8-His	1.05	0.84
	10 % pNEDD8-His	0.91	2.04
H302A	10% Ub-His	1.00	1.00
	10% pUb-His	0.98	1.90
	10% NEDD8-His	1.04	0.87
	10 % pNEDD8-His	0.91	1.30

Table 2 for reviewer. Quantification of the results shown in Fig. 1d.

As an alternative, the authors use western blots with anti-ubiquitin but there are a couple of issues that cast doubt on their reliability. Firstly, the westerns show significant ligase activity in the absence of either pUb or pNEDD8. Secondly, it is very odd that the authors load twice as much material on the gel for the western blots as for Coomassie staining. My interpretation is that a small amount of Parkin in their samples is partially denatured and responsible for the activity without pUb. Parkin activity can be partially activated simply by mild heating (<https://pubmed.ncbi.nlm.nih.gov/23861750/>). The absence of changes in the Coomassie gels suggests that only a small fraction of the Parkin molecules are activated in the presence of pNEDD8. The bottom line is that the evidence for Parkin activation by pNEDD8 is not compelling.

Firstly, as discussed above, we are using an N-terminally tagged version of Parkin and fusion of an N-terminal tag was reported to increase the basal activity of Parkin. More importantly, addition of pNEDD8 or pUb still results in significant stimulation of Parkin. Secondly, we thank the reviewer for spotting this mistake, which has been corrected (1/3 was analyzed by Western blot analysis, 2/3 by Coomassie staining).

We can, of course, not exclude that a small amount of partially denatured Parkin was present in our preparation, but if so, this most certainly was not due to "mild heating". In fact, the study cited by the reviewer treated Parkin at 56 °C with a T_m of Parkin of 55 °C. We would not consider this as "mild heating", but again, the main issue is that we observe prominent activation of Parkin by pUb and to a lesser extent by pNEDD8.

The phosphoserine binding sites on Parkin are well-known and mutations that block activation by pUb and pUbl (H302A or K211N) are well characterized. To validate their Parkin autoubiquitination assays, the authors could test activation with these Parkin mutants. The results would prove that pNEDD8 is working through direct binding of Parkin and have the additional value of potentially identifying the binding site.

Thank you for this suggestion. Accordingly, we performed ubiquitylation assays with the H302A mutant. As shown in the new Fig. 1d, the activity of this mutant was neither stimulated by pUb nor by pNEDD8 indicating that pNEDD8 stimulates Parkin via direct binding.

The NMR evidence for two conformations of phosphorylated Ub and NEDD8 was generally satisfactory but the depth of the structural analysis was limited. I found it surprising that although the authors claim NEDD8 is a good substrate of PINK1, they needed to use genetic code-expansion to directly incorporate phosphoserine and obtain sufficient, homogeneously phosphorylated protein. However, in spite of this, the pNEDD8 NMR samples still contained significant amounts of unphosphorylated protein which limited the analysis to the interpretation of chemical shifts.

First, we would like to point out that we were using genetic code expansion since this method is much more convenient, at least for us, than kinase-mediated phosphorylation for obtaining highly concentrated isotopically enriched samples of pNEDD8 which permits the acquisition of highly resolved two-dimensional and even three-dimensional NMR spectroscopic experiments. We underline that we have been successful in the assignment of chemical shifts of more than 80 % of

residues comprising both the retracted and the relaxed state of pNEDD8 (Figs. 2a-d). Please note that the application of TALOS-N (Fig. 2e and Figs. S4a, b, and d) and CSI analysis (Fig. S4e) expands the structural analysis conducted of both the retracted and the relaxed state of pNEDD8 significantly compared to a sole use of chemical shift values. Moreover, we have successfully applied NMR ZZ exchange methodology to obtain insights into the dynamic equilibrium that exists between both states (Figs. S3). The rate of interconversion between the two states could be even quantitatively determined (Fig. S3).

We note that the purification procedure for pNEDD8 - regardless of the phosphorylation methodology that has been employed (enzymatic or genetic code expansion) - is more cumbersome than pUb purification (note that in general, NEDD8 is structurally less stable than Ub making purification more difficult). In more detail, anionic exchange chromatography did not work for pNEDD8. Thus, a complete separation between phosphorylated and non-phosphorylated species of NEDD8 was not possible causing cross-peaks belonging to the unmodified form, as seen in a two-dimensional ¹H-¹⁵N HSQC NMR spectrum. However, the large majority of backbone resonances comprising both the relaxed and retracted state of pNEDD8 has been successfully assigned permitting an extensive structural characterization of pNEDD8 by analyzing chemical shift perturbations, TALOS-N, and chemical shift indices. Please note that proton, nitrogen as well as carbon resonances have been used by us to conduct a profound structural analysis. All our NMR data show that key structural and dynamic features of pUb - namely (i) the β 5-strand slippage in the retracted conformation, and (ii) the dynamic exchange between retracted and relaxed conformation on the slow micro- to millisecond time scale - are conserved in pNEDD8. In conclusion, we respectfully disagree with the notion that the analysis to the interpretation of chemical shifts was limited.

The last section of the manuscript was the interactome analysis. This naturally generated a lot of data but the robustness and completeness of the results were undercut by the absence of Parkin among the hits for pUb binding. Parkin binds pUb with nanomolar affinity. The authors need to explain why it was absent.

The notion that Parkin binds pUb with nanomolar affinity is somewhat misleading. Our studies were performed with unphosphorylated Parkin, which has a reported affinity for pUb of 400-500 nM (refs. 9, 11 of the manuscript; Sauvé et al., EMBO J. 34, pp. 2492). Furthermore, in contrast to SH-SY5Y cells, Parkin is not expressed in HEK293T cells. Finally, as indicated by reviewer 3, our method for preparing cell lysates is likely to decrease the chance to identify mitochondria-associated proteins. Taken together, it seems not surprising that we did not identify Parkin among the significant hits in our pUb/pNEDD8 interactome analysis. In this context, we would like to emphasize that the aim of the interactome analysis was not to support the notion pUb/pNEDD8 is involved in Parkin-mediated mitophagy but to provide evidence that pUb is likely to have additional cellular functions.

The MS analysis identified numerous interactions of pNEDD8 with protein chaperones such as HSP70. The physiological relevance is again unclear but could be the result of partial unfolding or misfolding of pNEDD8. In agreement with chaperone binding, the authors observed stimulation of HSP70 ATP-activity by pNEDD8. But, it was hard to see how the interactome results were related

to the PINK1-Parkin pathway or how they were evidence of biological relevance of NEDD8 Ser65 phosphorylation.

Our NMR data clearly indicate that the way we prepare NEDD8/pNEDD8 results in properly folded proteins. Furthermore, Bailly et al. reported that a hydrophobic patch mutant of NEDD8 does not stimulate the ATPase activity of HSP70 indicating the ability of NEDD8 to stimulate HSP70 is not due to the notion that NEDD8 is a client protein for HSP70. Therefore, we generated hydrophobic patch mutants of NEDD8 and pNEDD8 and tested their ability to stimulate HSP70 in comparison to the wild-type forms. This clearly showed that the integrity of the hydrophobic patch of pNEDD8 is critical for its ability to stimulate HSP70 activity. Thus, although we can only speculate on the physiological role of this observation, it is clearly not explained by the notion that pNEDD8 is a client protein of HSP70. Finally, we assume that the reviewer agrees that protein-protein interactions are pivotal for the function of any protein. Thus, a careful interactome analysis represents an important step in understanding the biological functions of a post-translationally modified protein. To prove these within cells is notoriously difficult when studying posttranslational modifications. In the vast majority of cases, these cannot be influenced in cells at will, and with respect to phosphorylation, results obtained with proteins containing phosphomimetic amino acids may not always be informative (e.g. in the case of Ub and NEDD8, glutamate or aspartate do not resemble the effects of S65 phosphorylation on the structural level and thus likely also not on the functional level).

Minor points:

The use of the term ubiquitin-like modifier to refer to NEDD8 would be less confusing than "Ubl" which is also used for the Ubl domain of Parkin.

Thank you for pointing this out. To avoid confusion, the Ubl domain of Parkin has been changed to "Ub-like domain".

The volume of the autoubiquitination assay should be given so that the pUb and pNEDD8 concentrations can be calculated. The amounts are given as weights compared to molar concentrations for the other components.

Reaction volume and concentrations have been included

In Fig 4f, the HEK293T clusters 5 and 6 seem to be missing.

For clusters 5 and 6, none of the GO terms was enriched. This is now indicated in the legend to Fig. 5a (4f of the original manuscript).

Reviewer #3

In their study entitled "Structural and functional consequences of NEDD8 phosphorylation", the authors report and dissect the structural consequences of the phosphorylation of NEDD8, a Ub-like protein, in Ser65. Phosphorylated NEDD8 activates the Ub ligase Parkin. In addition, the authors study the interactomes of Ub, pUb, NEDD8, pNEDD8, and also of non-hydrolyzable pUb and pNEDD8. Analysis of these interactomes unveils novel cellular functions for pUb and pNEDD8 beyond Parkin activation and mitophagy. In particular, the authors found that pNEDD8 enhances the activity of HSP70 in vitro.

The experiments are well designed, the overall quality of the data is high, the information is clear, and the manuscript is well written.

Although describing very interesting findings as the phosphorylation of NEDD8, its role in Parkin activation, its structural consequences, and the differential interactome of the different forms of Ub and NEDD8, there are some aspects that, in my opinion, would require further investigation to be published in Nature Communications.

Major comments:

1. Lack of biology. You would expect some assays showing the biological consequences of NEDD8 phosphorylation in addition to the reported in vitro assays.

1.1. Ubiquitin is phosphorylated on Ser65 upon mitochondrial depolarisation leading to Parkin activation and mitophagy. I would suggest to the authors to induce mitochondrial depolarisation (i.e. with carbonyl cyanide 3-chlorophenylhydrazone (CCCP)) and look at presence of phosphorylated NEDD8 using the Phos-tag gels followed by Immunoblotting against NEDD8 (I do not think there is an antibody against pNEDD8, but Immunoblotting against it would definitely be a better option).

1.2. The authors suggest that DNA damage induces NEDD8 phosphorylation - they could do the same as in point 1.1. using DNA damage agents to prove this claim.

1.2. What are the consequences in Mitophagy after introducing a phosphodeficient mutant of NEDD8 (S65A) in NEDD8 KO cells? This model could be used to study the consequences of not having phosphorylated NEDD8 in the additional signaling pathways identified in the interactome analysis such as autophagy, DNA damage, apoptosome formation, etc.

Thank you for these helpful suggestions. Accordingly, we ectopically expressed an N-terminally His tagged version of NEDD8 in cells, treated these with CCCP to induce mitochondrial stress, and purified NEDD8 under denaturing conditions. While this did not allow to detect pNEDD8 on a Phos tag gel (possibly because pNEDD8 levels are too low), we were able to show by mass spec analysis that CCCP stimulates phosphorylation of NEDD8 by ~1.7-fold (which appears to be similar to published data on Ub phosphorylation. In addition total levels of pNEDD8 appear to be similar to pUb; ref. 11 of the manuscript). In addition, the mass spec analysis allowed us to analyze the phosphorylation status of NEDD8 specifically at S65. These data are included in the revised manuscript as Fig. 1h and in Fig. S1d-e.

We also determined the phosphorylation status of NEDD8 upon treatment of cells with the DNA damaging agent etoposide, which mainly introduces DNA double strand breaks (via inhibition of topoisomerase II). Intriguingly, this showed that in contrast to mitochondrial stress, this particular type of DNA damage results in a decrease of NEDD8 phosphorylation (Fig. S6e-g). We discuss this new data in the manuscript in the context of the effect of pNEDD8 on Hsp70 activity (page 17, 1st paragraph). Although we can only speculate on the functional consequences of this differential phosphorylation, we are convinced that the notion that different stress stimuli have different effects on NEDD8 phosphorylation are of significant interest to the community.

Due to its critical role in the regulation of Cullin-dependent ubiquitin ligase complexes, it is not possible to generate NEDD8 KO cells. Alternatively, NEDD8 could be downregulated by RNA interference. However, Ub would still be present. Since it would not be possible to simultaneously knockdown Ub expression and NEDD8 expression (as in contrast to NEDD8, Ub is not only involved in Parkin activation but is also required for ubiquitylation), we believe that results obtained in NEDD8 knockdown experiments would not be informative or rather difficult to interpret. Although we cannot draw conclusions on how much NEDD8 phosphorylation contributes to Parkin activation, we hope that the reviewer agrees that the observation that NEDD8 phosphorylation is stimulated by mitochondrial stress at least suggests that it is involved. In other words, we do not claim that pNEDD8 is more or similarly important for Parkin activation as pUb; we just indicate that pNEDD8 has the potential to contribute to it.

2. Validation of the interactome data using an orthogonal method such as Immunoblotting after enrichment with the described baits or, alternatively, performing Co-Immunoprecipitations using existing Ub, phospho-Ub, and NEDD8 antibodies.

As suggested, the interactome data were validated by Western blot analysis for some of the proteins demonstrating the robustness of our mass spec analysis. The data are presented in the new Fig. 4f.

Minor points:

1. Can you prove that the NEDD8-His variant does not conjugate to proteins using an in vitro NEDDylation assay?

As suggested, we performed in vitro neddylation experiments. This clearly showed that as expected, NEDD8-His cannot be used for conjugation to proteins (Fig. 4 for reviewer). We hope that the reviewer agrees that these data are of limited interest and should not be incorporated in the manuscript.

Fig. 4 for reviewer. In vitro neddylation assay was performed with Ub/NEDD8 variants in the presence of the NEDD8-activating enzyme and NEDD8 E2s as indicated. Reactions were stopped after 60 min and analyzed by SDS-PAGE followed by Coomassie blue staining. Running positions of molecular mass markers, the Ub/NEDD8 variants, the E2s, and auto-neddylated forms the E2s are indicated.

2. To preserve mitochondrial proteins the centrifugation speed during cell lysis should be lower than the 21,000 g the authors used here. This might be the reason the usual suspects were not identified in this interactome study (Parkin, USP30, and other mitochondrial proteins such as VDACS).

Thank you for this information. We introduced a respective sentence in the Methods section of the revised manuscript (page 26, 1st paragraph).

3. A more elaborate description of the stress pathways identified would be beneficial for the quality of the manuscript.

As suggested, we have added some additional information (page 13/14, last/1st paragraph; page 14, 2nd paragraph; page 17, 1st paragraph but due to space constraints, we did not go into much detail. However, we will be happy to do so, if the reviewer feels this to be important.

REVIEWER COMMENTS

Reviewer #1 (Remarks to the Author):

The authors have addressed the major points raised in the previous version of the manuscript. I think the demonstration of the response of NEDD8 phosphorylation upon different stress conditions is important as it indicates a regulatory role for this modification. Additionally, they have strengthened the biochemical analysis on the Parkin activation.

Based on the recent NEDD8 literature the authors may find useful to discuss the study by Kim et al., 2021 *iScience* on HDAC6 regulation by NEDD8 and include in their introduction the NEDD8 proteomics study by Lobato et al., 2021, *Cell Reports* along with the Vogl et al. 2020, *Nat Struc.Mol. Biol.* study.

Reviewer #2 (Remarks to the Author):

This is an improved manuscript and the authors have addressed many of the issues in the previous manuscript. That said, I'm still not convinced that *Nature Communications* is the proper venue. The results are largely descriptive and the biological relevance of NEDD phosphorylation is untested. The activation of parkin looks to be real but the effect is small and not supported by other data. The structural changes in NEDD8 upon phosphorylation are nicely characterized but of unclear physiological significance. The enrichments shown in the MS results appear limited in size. Finally, the effect of pNEDD8 activation of HSP70 is rather small and again of unclear biological significance. To be clear, the results are worthy of publication and, once revised, the manuscript will be a useful addition to the literature, but I think that it is more suitable for more specialized journal.

Below, I've listed some suggestions to help the authors.

- 1) The gel quantifications provided in the rebuttal letter should be included in the main figures. For example, the percent phosphorylation in "Reviewer Table 1" should be added to the bottom of figure 1b.
- 2) Similarly, there is no reason not to include some of the "Reviewer Figures", such as the parkin RORBR activity assay as supplemental figures.
- 3) Most journals will require plots that show the individual data points rather than just the average and SEM (e.g. in Fig 1h or S6d).
- 4) While the authors state that the activation of parkin is not the key result of paper, it nonetheless remains the title of the first figure. The activation experiment is much improved. The inclusion of the parkin H302A mutant is a welcome addition and suggests that the effect is real. However, the level of activation remains much lower than activation by pUb, which itself is much lower than the activation by phosphorylation of parkin (for comparison see Fig 5F of Sauvé et al., *EMBO J.* 34, pp. 2492 and the structure papers from Gladkova et al., *Nature* 559, pp. 410 and Sauvé et al., *NSMB* 25, pp. 623.) The weak activation by pNEDD8 might be of interest if the paper addressed the mechanism or biological significance of the effect but it doesn't.
- 5) Fig 1d & e, it would be helpful if the figure or figure legend indicated the concentration of Ub/NEDD8 added (in addition to the variant ratios shown in 1e).
- 6) The level of NEDD8 phosphorylation is quite small, which isn't a problem on its own, but it doesn't increase significantly upon CCCP treatment, which is more problematic. The authors claim that the levels change 1.7-fold but that seems to be measurement noise rather than a real effect. The non-phosphorylated peptide from NEDD8 S65A mutant shows large variations (Fig S1e and S6g) as do the control peptide in the Excel worksheet. For example, the ratio between the NEDD8 peptides

EIEIDIEPTDK and EGIPPQQQR varied between 0.75 and 1.7 in different experiments.

6) It wasn't clear how the authors used the values in the worksheet to calculate the values in the Table S1e but, as a simple alternative, I calculated the fraction of phosphorylated NEDD8 from the values for the non-phosphorylated and phosphorylated peptide (ILGGSVLHLVLALR). Including the results with etoposide, I get:

Minus CCCP 1.08%
Minus CCCP 0.79%
Plus CCCP 2.51%
Plus CCCP 1.00%
Minus Etoposide 2.05%
Minus Etoposide 1.60%
Plus Etoposide 1.20%
Plus Etoposide 1.31%

Unfortunately, I don't see a trend. Perhaps the authors would see a larger effect with a sample enriched for mitochondrial proteins or some other cellular fraction but, in their current form, the results in the paper do not show clear changes in the levels of NEDD8 phosphorylation from either CCCP or etoposide treatment.

8) Fig S1b, it would be helpful if the horizontal axes were expanded or a zoom provided so that the degree of purity of the proteins could be seen. The authors state that there was some unphosphorylated NEDD8 in the purified sample but it isn't visible.

9) I had difficulty understanding the MS heat plots, specifically the meaning of the "beads only" column. The Excel datasheet didn't show the values or how the enrichments were calculated. Are the enrichments calculated relative to the average for all conditions? It is odd that the Excel datasheet does not include a summary column showing the calculated enrichment and p-value.

10) The scale of the heat plots (-2 to 2) is too small which makes it difficult to compare values. For example, the text on page 12 states "For the deneddylating enzymes identified, COPS5, COPS6, OTUB2 and SENP8 preferentially bound to unmodified NEDD8 (Fig. 5b)," but the heat plot in Fig 5b shows red for both NEDD8 and pNEDD8. Why do the proteins bind pNEDD8 but not p(nh)NEDD8?

11) A volcano plot would be a better way to allow comparison between individual proteins.

12) The paper never gives specific values or statistics for the enrichments. For example, how does HSP70 (HSPA1B) compare with, for example, the enrichment of UBE2M? or HDAC6?

13) What is the enrichment of HSP70 binding by p(nh)NEDD8 over pNEDD8? The text says "strongly" but I only see a 2- to 3-fold effect in the MS data.

14) Why is HSP70 (HSPA1B) missing from the western in Fig 4f? Given its significance for the story, it is an unfortunate omission.

15) The points raised by the authors about the difficulty of detecting parkin in cell extracts are well-taken but don't explain why the authors don't show pulldowns of recombinant parkin and HSP70 by pNEDD8.

16) The stimulation of HSP70 ATPase activity by pNEDD8 needs to be better characterized. The addition of data with a hydrophobic patch (LIA) mutant of NEDD8 is nice but the experiments are single endpoint assays and don't report absolute rates. The authors need to show time courses with absolute rates. The Materials & Methods section states that the assay was "started by the addition of

0.5 μ l of 11 mM ATP to reach a final reaction volume of 30 μ l." Given the difficulty of pipetting accurately 0.5 μ l, the authors should verify that this is not a potential source of variability in the results (i.e. is 180 μ M much larger than the K_m ?) Lastly, the low ATPase activity of Hsp70 proteins makes the possibility of contaminant protein with high ATPase activity a concern. Did the authors include controls without HSP70?

These may appear to be pedantic concerns but the HSP70 results show significant variability. Fig S6d shows pNEDD8 leads to a 2.5-fold enhancement in ATPase activity but Fig 6d shows only 1.8-fold. Is this due to a difference in the buffer rate or pNEDD8 rate? Is the 1.6-fold enhancement by unphosphorylated NEDD8 in Fig 6d significant?

In summary, the paper is much improved and wherever it is published, the revisions have been worthwhile. While it convincingly identifies an in vitro effect of pNEDD8 on parkin activation, two pNEDD8 conformations similar to those observed with pUb, and the interactomes of pUb and pNEDD8, it still doesn't show a biological role of NEDD8 phosphorylation in either parkin or chaperone activity.

Reviewer #3 (Remarks to the Author):

I would like to thank the authors for the effort to reply to my points.

Most of my concerns have been satisfactorily allayed.

However, I still believe that some more functional studies could be done to show the biological relevance of NEDD8 phosphorylation. For example: stably transfecting NEDD8 (containing an alternative sequence of codons to generate the protein) deficient for phosphorylation, and subsequently knocking out the endogenous NEDD8 would be possible and it would allow testing the consequences of preventing the phosphorylation of NEDD8 in certain cellular responses such as mitochondrial depolarisation, mitophagy, and DNA damage.

Point-by-point response to reviewers' comments on "Structural and functional consequences of NEDD8 phosphorylation" (NCOMMS-20-39525A)

Reviewer #1:

The authors have addressed the major points raised in the previous version of the manuscript. I think the demonstration of the response of NEDD8 phosphorylation upon different stress conditions is important as it indicates a regulatory role for this modification. Additionally, they have strengthened the biochemical analysis on the Parkin activation. Based on the recent NEDD8 literature the authors may find useful to discuss the study by Kim et al., 2021 iScience on HDAC6 regulation by NEDD8 and include in their introduction the NEDD8 proteomics study by Lobato et al., 2021, Cell Reports along with the Vogl et al. 2020, Nat Struc.Mol. Biol. study.

We thank the reviewer for the positive comments. According to the suggestions of the reviewer, the results by Kim et al. are briefly discussed on page 14 of the revised manuscript and the study by Vogl et al. is now referenced.

Reviewer #2:

This is an improved manuscript and the authors have addressed many of the issues in the previous manuscript. That said, I'm still not convinced that Nature Communications is the proper venue. The results are largely descriptive and the biological relevance of NEDD phosphorylation is untested. The activation of parkin looks to be real but the effect is small and not supported by other data. The structural changes in NEDD8 upon phosphorylation are nicely characterized but of unclear physiological significance. The enrichments shown in the MS results appear limited in size. Finally, the effect of pNEDD8 activation of HSP70 is rather small and again of unclear biological significance. To be clear, the results are worthy of publication and, once revised, the manuscript will be a useful addition to the literature, but I think that it is more suitable for more specialized journal. Below, I've listed some suggestions to help the authors.

We were glad to learn that the reviewer found our manuscript to be improved and are grateful for the suggestions that helped us to further strengthen our study.

1) The gel quantifications provided in the rebuttal letter should be included in the main figures. For example, the percent phosphorylation in "Reviewer Table 1" should be added to the bottom of figure 1b.

The respective numbers are now provided in Fig. 1b and 1d of the revised manuscript.

2) Similarly, there is no reason not to include some of the "Reviewer Figures", such as the parkin R0RBR activity assay as supplemental figures.

The Parkin Δ Ubl activity assay has been included as Supplementary Fig. 2c.

3) Most journals will require plots that show the individual data points rather than just the average and SEM (e.g. in Fig 1h or S6d).

We apologize for not having done this in the first place. The individual data points have been

included in Fig. 6d and Supplementary Figs. 7d and 7e. For Fig. 1h and Supplementary Fig. S7h, all replicates are now depicted separately.

4) While the authors state that the activation of parkin is not the key result of paper, it nonetheless remains the title of the first figure. The activation experiment is much improved. The inclusion of the parkin H302A mutant is a welcome addition and suggests that the effect is real. However, the level of activation remains much lower than activation by pUb, which itself is much lower than the activation by phosphorylation of parkin (for comparison see Fig 5F of Sauv   et al., EMBO J. 34, pp. 2492 and the structure papers from Gladkova et al., Nature 559, pp. 410 and Sauv   et al., NSMB 25, pp. 623.) The weak activation by pNEDD8 might be of interest if the paper addressed the mechanism or biological significance of the effect but it doesn't.

We are grateful to the reviewer for reminding us that Parkin activity is stimulated by at least two mechanisms, PINK1-mediated phosphorylation and interaction with pUb (or pNEDD8, as our data suggests) with phosphorylation of Parkin having a much stronger effect on Parkin activity than the interaction of non-phosphorylated Parkin with pUb. Importantly, for instance Wauer et al. (Nature 524, pp. 370) have shown that binding of pUb greatly increases the efficiency of PINK1-mediated phosphorylation of the Ub-like domain of Parkin. In the revised manuscript, we therefore tested whether similar to pUb, pNEDD8 stimulates PINK1-mediated phosphorylation of Parkin. Indeed, as shown in the new Fig. 1c, addition of pNEDD8 results in quantitative phosphorylation of Parkin, while addition of NEDD8 had no stimulating effect. Thus, just like pUb, binding of pNEDD8 has a two-fold effect on Parkin activity. It stimulates Parkin activity (i) indirectly by driving it into a conformation that is more accessible/susceptible to PINK1, and (ii) directly, though with 2-3 fold reduced efficiency compared to pUb (see Fig. 1d). Thus, we hope that the reviewer agrees that this new observation significantly strengthens the notion that pNEDD8 acts as an activator of Parkin (see also response to comment 15).

5) Fig 1d & e, it would be helpful if the figure or figure legend indicated the concentration of Ub/NEDD8 added (in addition to the variant ratios shown in 1e).

The respective concentrations are now included in the figure legend of Fig. 1d and e.

6) The level of NEDD8 phosphorylation is quite small, which isn't a problem on its own, but it doesn't increase significantly upon CCCP treatment, which is more problematic. The authors claim that the levels change 1.7-fold but that seems to be measurement noise rather than a real effect. The non-phosphorylated peptide from NEDD8 S65A mutant shows large variations (Fig S1e and S6g) as do the control peptide in the Excel worksheet. For example, the ratio between the NEDD8 peptides EIEIDIEPTDK and EGIPPQQR varied between 0.75 and 1.7 in different experiments.

We agree with the reviewer that there are variations in the peptide peak area for the non-phosphorylated peptide and the control peptides between the different biological replicates (see also below, response to comment 7). In fact, this was the reason for including the control peptides and the non-phosphorylated peptide in the measurements, as this allows to correct for variations between different samples within a given biological replicate. To do so, we cannot compare the absolute values of the peptide area between samples, but have first to calculate the ratio of phosphorylated vs. non-phosphorylated peptide within one sample

and then compare this to the ratio of other samples. Nonetheless, we agree that the ratios between the different biological replicates differ (see also below, response to comment 7) and, thus, to corroborate that CCCP treatment results in increased phosphorylation of NEDD8, we performed three additional biological replicates. The data are summarized in the new Fig. 1h.

7) It wasn't clear how the authors used the values in the worksheet to calculate the values in the Table S1e but, as a simple alternative, I calculated the fraction of phosphorylated NEDD8 from the values for the non-phosphorylated and phosphorylated peptide (ILGGSVLHLVLALR). Including the results with etoposide, I get:

Minus CCCP 1.08%

Minus CCCP 0.79%

Plus CCCP 2.51%

Plus CCCP 1.00%

Minus Etoposide 2.05%

Minus Etoposide 1.60%

Plus Etoposide 1.20%

Plus Etoposide 1.31%

Unfortunately, I don't see a trend. Perhaps the authors would see a larger effect with a sample enriched for mitochondrial proteins or some other cellular fraction but, in their current form, the results in the paper do not show clear changes in the levels of NEDD8 phosphorylation from either CCCP or etoposide treatment.

As abovementioned, we have included the data of 3 more biological replicates (CCCP treatment), and the normalized peak area values for the 5 biological replicates are depicted separately in Fig. S1e. The values were calculated in two steps. First, the peptide area of the non-phosphorylated and the phosphorylated peptide was each divided by the mean of the control peptides. Then, the sum of the values for the non-phosphorylated peptide and the phosphorylated peptide was set to 100% and the percentage of either peptide was calculated. For the values shown in Fig. 1h, the normalized peak area values for both conditions (DMSO/CCCP) of one biological replicate were divided by those of the DMSO control to set DMSO controls to 1 and depicting the fold change for the respective CCCP-treated samples. The same holds true for Fig. S7h and i (etoposide treatment).

Since we consistently observe an increase in NEDD8 phosphorylation upon CCCP treatment, though the increase varies between different biological replicates from 1.3 to 2.3-fold, we feel that it is fair to conclude that CCCP treatment induces NEDD8 phosphorylation. In this context, we would like to note that we deliberately used non-synchronized cells and did not check the state of confluency, when cells were lysed. Furthermore, each biological replicate represents a separate transfection experiment (for ectopic expression of NEDD8). Thus, variations between different biological replicates are kind of expected. Furthermore, as indicated in the first reviewing around, the fold change appears to be similar to that published for ubiquitin phosphorylation, and, moreover, biological replicates appear to be missing or show larger variations as well (Ordureau et al., Mol Cell 56, pp. 360; Lai et al., EMBO J 34, pp. 2840).

8) Fig S1b, it would be helpful if the horizontal axes were expanded or a zoom provided so that the degree of purity of the proteins could be seen. The authors state that there was some unphosphorylated NEDD8 in the purified sample but it isn't visible.

The degree of impurity with respect to unphosphorylated NEDD8 is indeed rather small and cannot be unambiguously identified in the ESI-MS spectrum (in fact, a peak in the respective area is not detectable). However, in the Phos-tag gel of Fig. S2a, a faint band at the running position of unphosphorylated NEDD8 is visible (that is why we concluded that there is some in the pNEDD8 sample). Based on this, we can safely conclude that the purity of pNEDD8 is >95 percent (and, thus, the amount of unphosphorylated NEDD8 present in the prep is neglectable).

9) I had difficulty understanding the MS heat plots, specifically the meaning of the "beads only" column. The Excel datasheet didn't show the values or how the enrichments were calculated. Are the enrichments calculated relative to the average for all conditions? It is odd that the Excel datasheet does not include a summary column showing the calculated enrichment and p-value.

As indicated in the legend to Fig. 4a, the "beads only" column represents empty beads, which were used as control to exclude proteins that bind unspecifically to the matrix. To do so, the "empty beads" control was handled in a manner identical to the beads decorated with bait molecules (incubation with cell extract, affinity enrichment, digestion, and LC-MS/MS).

We are not sure what is meant by the statements "*The Excel datasheet didn't show the values or how the enrichments were calculated, etc.*". The first two sheets of the Source Data show for all significantly enriched proteins the calculated values for the average enrichment (log₂, Z-scored, with Z-score defined as the mean of each row is subtracted from each value and the result is divided by the standard deviation of the row), the LFQ intensities (log₂, Z-scored), and ANOVA statistics (p-value). As mentioned in the Methods section, the significantly enriched proteins were identified by ANOVA, the rows were Z-score normalized, and the replicates were averaged, leading to the values that are the basis for the hierarchically clustered heatmaps shown in Fig. 4b. The Source Data sheet additionally shows the respective clusters, GO Annotations as well as all identified proteins (without any filtering).

10) The scale of the heat plots (-2 to 2) is too small which makes it difficult to compare values. For example, the text on page 12 states "*For the deneddylating enzymes identified, COPS5, COPS6, OTUB2 and SENP8 preferentially bound to unmodified NEDD8 (Fig. 5b), but the heat plot in Fig 5b shows red for both NEDD8 and pNEDD8. Why do the proteins bind pNEDD8 but not p(nh)NEDD8?*"

We respectfully note that that the values (-2 to 2) are log₂ transformed, which is commonly used to present respective data. In fact, the scale of hierarchically clustered heatmaps generated with Perseus is specified by the program itself and displays the optimal range covering all presented values. For comparison of the original (untransformed) values, we refer to the Source Data file sheet 5 and 6.

On page 16/17 of the manuscript, we discuss possible reasons for differences in the interactomes between natively phosphorylated pUb/pNEDD8 and their non-hydrolyzable

variants. In case of COPS5, COPS6, OTUB2 and SENP8, a likely explanation for the different binding behavior of pNEDD8 and ^{nh}pNEDD8 is provided on page 17: "As abovementioned, another possibility is that the natively phosphorylated Ub/NEDD8 variants are partly dephosphorylated during incubation with whole cell lysates. The resulting mixture of phosphorylated and unphosphorylated forms of Ub/NEDD8 will for obvious reasons result in a different pattern of interactors compared to the non-hydrolyzably phosphorylated Ub and NEDD8 variants."

11) A volcano plot would be a better way to allow comparison between individual proteins.

A volcano plot is suited to compare the behavior of individual proteins between two conditions. However, we are comparing the interaction of individual proteins with 6 different bait molecules and the empty beads control (i.e. 7 conditions). Thus, ANOVA statistics in combination with hierarchically clustered heatmaps are - in our opinion - much better suited than volcano plots to visualize different enrichment patterns for the baits used.

12) The paper never gives specific values or statistics for the enrichments. For example, how does HSP70 (HSPA1B) compare with, for example, the enrichment of UBE2M? or HDAC6?

We have not included specific values for the enrichments in the text of the manuscript, but these as well as statistics are presented in the Source Data file sheet 1 and 2. For the HEK293T data set, for example, the values for HSPA1B, UBE2M, and HDAC6 are indicated in line 25, 29, and 67, respectively, in sheet 1 ("1_Fig. 4b_HEK293"). The values for these examples are the following:

		Average Enrichment (Log2, Z-scored)							
Clu	ster	Beads	Nedd8 nhpS65	Nedd8 pS65	Nedd8 Nedd8	Ub nhpS65	Ub pS65	Ub Ub	-Log ANOVA p value
		HSPA1B; HSPA1A	2	-0.52585	1.68459	0.89644	0.650308	-0.85237	-0.78724
UBE2M	3	-0.85344	1.05964	1.12106	1.05725	-0.87185	-0.26785	-1.16706	21.4984
HDAC6		-2.24408	0.29976	0.47090	-0.09998	0.660299	0.794479	0.165681	19.6441

13) What is the enrichment of HSP70 binding by p(nh)NEDD8 over pNEDD8? The text says "strongly" but I only see a 2- to 3-fold effect in the MS data.

As abovementioned (comment 10), it needs to be considered that the values are log2 transformed. Thus, a 2- to 3-fold effect on the log2 scale translates into a 4- to 8-fold effect on a linear scale (which we consider a strong effect). The values for this example (Source data file, sheet 1, line 24) are:

		Average Enrichment (Log2, Z-scored)							
Gene names	Clu ster	Beads	Nedd8 nhpS65	Nedd8 pS65	Nedd8 Nedd8	Ub nhpS65	Ub pS65	Ub Ub	-Log ANOVA p value
		HSPA8	2	-0.36291	1.97833	0.72232	0.398432	-0.79949	-0.73295

14) Why is HSP70 (HSPA1B) missing from the western in Fig 4f? Given its significance for the story, it is an unfortunate omission.

The HSP70 family consists of a number of proteins including HSPA1A/B, HSPA4, HSPA4L, HSPA8, and HSPA9, which were identified to preferentially interact with phosphorylated NEDD8 in our study (Fig. 6a-c; note that HSPA1B and HSPA8 are 85% identical and 93% similar at the amino acid sequence level). Due to the availability of an anti-HSPA8 antibody and an HSPA8 cDNA in our lab, we have chosen HSPA8 as a representative of the HSP70 family members for the western blot verification shown in Fig. 4f. Importantly, for the ATPase activity assay we also used the ATPase domain of HSPA8 (residues 1-402, which included the ATPase activity stimulating linker domain). This is indicated in the Methods section on page 18. To avoid any confusion, this information has also been added to the main text and the legend of Fig. 6d.

15) The points raised by the authors about the difficulty of detecting parkin in cell extracts are well-taken but don't explain why the authors don't show pulldowns of recombinant parkin and HSP70 by pNEDD8.

We thank the reviewer for this helpful comment. To address this issue, we performed a pull-down assay with GST fusion proteins of wild-type Parkin, Parkin Δ Ubl, and Parkin H302A (new Fig. 1f). Importantly, the results obtained correlate well with the data obtained in autoubiquitylation assays and further strengthen the notion that pNEDD8 acts as a stimulator of Parkin activity. For example, while pNEDD8 bound to Parkin and Parkin Δ Ubl, an interaction with the H302A mutant was not observed.

Since HSPA8 (as representative of HSP70 family members) was included in the western blot verification shown in Fig. 4f, we hope that the reviewer agrees that pull-down assays with recombinant "HSP70" would not add new information.

16) The stimulation of HSP70 ATPase activity by pNEDD8 needs to be better characterized. The addition of data with a hydrophobic patch (LIA) mutant of NEDD8 is nice but the experiments are single endpoint assays and don't report absolute rates. The authors need to show time courses with absolute rates. The Materials & Methods section states that the assay was "started by the addition of 0.5 μ l of 11 mM ATP to reach a final reaction volume of 30 μ l." Given the difficulty of pipetting accurately 0.5 μ l, the authors should verify that this is not a potential source of variability in the results (i.e. is 180 μ M much larger than the K_m ?) Lastly, the low ATPase activity of Hsp70 proteins makes the possibility of contaminant protein with high ATPase activity a concern. Did the authors include controls without HSP70?

A time course experiment is now shown in Supplementary Figure S7e and the absolute values of ATP consumption are provided in Fig. S7f (the values are provided in μ M rather than a rate, as this makes it easier to judge how much of the initial ATP concentration was used up). The experimental setup was adapted from Bailly, et al. (Cell Rep. 29, pp. 212). Furthermore, 180 μ M is much larger than the reported K_m value of HSP70 (1.37 μ M; Sadis S & Hightower LE. Biochemistry 31, pp. 9406-9412) and, thus, inaccurate pipetting is unlikely a source of variability. As mentioned in the Methods section, controls without HSPA8 were performed to correct for non-enzymatic ATP hydrolysis (raw data and calculated values are provided in the Source Data file, sheet 9).

These may appear to be pedantic concerns but the HSP70 results show significant variability. Fig S6d shows pNEDD8 leads to a 2.5-fold enhancement in ATPase activity but Fig 6d shows only 1.8-fold. Is this due to a difference in the buffer rate or pNEDD8 rate? Is the 1.6-fold enhancement by unphosphorylated NEDD8 in Fig 6d significant?

Determination of the ATPase activity was performed with the malachite green reagent as an indirect read out for ATP hydrolysis (i.e. the free phosphate generated during ATP hydrolysis is detected) in 96-well plates. Thus, the variation in HSPA8 activity likely originates from the malachite green detection reaction rather than actual differences in HSPA8 activity: depending on the number of samples of the individual experiments, the time it takes to add the reaction mixture to malachite green in the individual wells of the 96-well plate differs. In other words, the starting point of the malachite green reaction and thus the exact time of the absorbance measurement are not exactly the same between different experiments. This inaccuracy results in some variability, but since within an individual experiment we always observe the same trend, we do not consider this to be of concern.

In summary, the paper is much improved and wherever it is published, the revisions have been worthwhile. While it convincingly identifies an in vitro effect of pNEDD8 on parkin activation, two pNEDD8 conformations similar to those observed with pUb, and the interactomes of pUb and pNEDD8, it still doesn't show a biological role of NEDD8 phosphorylation in either parkin or chaperone activity.

Reviewer #3:

I would like to thank the authors for the effort to reply to my points.

Most of my concerns have been satisfactorily allayed.

However, I still believe that some more functional studies could be done to show the biological relevance of NEDD8 phosphorylation. For example: stably transfecting NEDD8 (containing an alternative sequence of codons to generate the protein) deficient for phosphorylation, and subsequently knocking out the endogenous NEDD8 would be possible and it would allow testing the consequences of preventing the phosphorylation of NEDD8 in certain cellular responses such as mitochondrial depolarisation, mitophagy, and DNA damage.

We agree with the reviewer that additional experiments will be required to unambiguously demonstrate the biological relevance of NEDD8 phosphorylation. However, the experiment suggested by the reviewer will take a considerable amount of time. Moreover, ectopic expression of a NEDD8 mutant may affect cell viability or other cellular properties/pathways. Thus, to obtain clear-cut results, it may even be necessary to employ an inducible system for NEDD8 expression. We hope that the reviewer agrees that even in the absence of such experiment, our data in sum make a good case for the biological relevance of NEDD8 phosphorylation.